

# Design and Performance of the Hotrod Melt-Tip Ice-Drilling System

William Colgan[1], Christopher Shields[1], Pavel Talalay[2], Xiaopeng Fan[2], Austin P. Lines[3], Joshua Elliott[3], Harihar Rajaram[4], Kenneth Mankoff[1,7], Morten Jensen[5], Mira Backes[6], Yuenchen Liu[2], Xianzhe Wei[2], Nanna B. Karlsson[1], Henrik Spanggård[1] and Allan Ø. Pedersen[1]

[1]Geological Survey of Denmark and Greenland, DENMARK
[2]Polar Research Center, Jilin University, CHINA
[3]Polar Research Equipment, USA
[4]John Hopkins University, USA
[5]Copenhagen School of Design and Technology, DENMARK
[6]The Technical University of Denmark, DENMARK
[7]National Snow and Ice Data Center, University of Colorado Boulder, USA

*Correspondence to*: William Colgan (wic@geus.dk)

**Abstract.** We introduce the design and performance of a melt-tip ice-drilling system designed to insert a temperature sensor cable into ice. The melt tip is relatively simple and low cost, designed for a one-way trip to the ice-bed interface. The drilling

system consists of a melt tip, umbilical cable, winch, interface, power supply, and support items. The melt tip and the winch are the most novel elements of the drilling system, and we make the hardware and electrical designs of these components available open access. Tests conducted in a laboratory ice well indicate that the melt tip has an electrical energy to forward melting heat transfer efficiency of ~35% with a theoretical maximum penetration rate of ~12 m/hr at maximum 6.0 kW power. In contrast, ice-sheet testing suggests the melt tip has an analogous heat transfer efficiency of ~15% with a theoretical

maximum penetration rate of ~6 m/hr. We expect the efficiency gap between laboratory and field performance to decrease with increasing operator experience. Umbilical freeze-in due to borehole refreezing is the primary depth-limiting factor of the drilling system. Enthalpy-based borehole refreezing assessments predict refreezing below critical umbilical diameter in ~4 hours at -20 ˚C ice temperatures and ~20 hours at -2 ˚C. This corresponds to a theoretical depth limit of up to ~200 m, depending on firn thickness, ice temperature and operator experience.

## 1 Introduction

The thermal state of the ice-bed interface is a critical boundary condition for understanding the form and flow of an ice sheet. Geothermal heat flow provides the basal boundary condition for the thermodynamics in an ice-sheet model. The presence or absence of basal sliding similarly provides the basal boundary condition for the continuum mechanics in an ice-sheet model. Presently, however, there is poor scientific agreement over whether the ice-bed interface is at, or below, pressure-melting-

point temperature beneath an estimated one-third of the Greenland Ice Sheet [MacGregor et al., 2022]. There is also substantial disagreement between regional models of geothermal heat flow across Greenland, which approaches 100% relative disagreement in South Greenland [Colgan et al., 2021].



While many boreholes have been drilled around the ice-sheet periphery, basal temperature and geothermal heat flow have only been directly sampled in the ice-sheet interior at six sites in the last six decades [Løkkegaard et al., 2022]. These six sites
denote the locations of the deep Greenland ice cores: Camp Century (1966), DYE-3 (1981), GISP2 (1993), GRIP (1998), NGRIP (2003) and NEEM (2010). Retrieving each of these invaluable ice core records represents a tremendous multi-annual logistical and scientific undertaking [Langeway, 2008]. At these sites, ice samples are collected and analysed as the primary data. Basal temperature and geothermal heat flow are measured as secondary data. Demand for increasingly detailed prognostic simulations of ice-sheet form and flow, however, now provides a strong impetus to drill deep boreholes for the primary purpose
of measuring basal thermal state [Siegert et al., 2020].

Here, we describe the design and performance of a prototype ice drilling system that has the sole purpose of rapidly deploying thermistor strings to the ice-bed interface with minimum logistical support. The fundamental concept is to pull a thermistor cable into the ice sheet behind a compact and inexpensive melt tip on a one-way trip to the ice-bed interface. We describe the laboratory and field testing of this ice-drilling system. We also discuss the efficiency and applicability of our
drilling system in the context of the rich history of melt-tip ice drills [Talalay, 2019]. Finally, following open-science best practice, we have released the computer aided design (CAD) schematics, machining specifications and source code associated with this drilling system at https://doi.org/10.22008/FK2/DXXR06 [Colgan et al., 2022], in the hope of further accelerating improvements in melt-tip technology.

## 2 Design

Below, we describe the drilling system in six sections: Melt Tip (Section 2.1), Umbilical Cable, (Section 2.2), Winch (Section 2.3), Interface (Section 2.4), Power Supply (Section 2.5), and Support Items (Section 2.6).

### 2.1 Melt Tip

The primary function of the melt tip is to transfer heat from its internal cartridge heaters into the ice below the melt tip as efficiently as possible. The melt tip therefore seeks to convert electrical energy and dissipate the resulting heat flux in a down-
borehole direction. As it is possible for the melt tip to reach internal temperatures >400 °C, ensuring that the melt-tip components, both structural and electrical, can operate over an extreme temperature range (-40 to +400 °C) presents an appreciable design challenge. The main components of the melt tip are a copper heating block, an electronics package, and structural members. The melt tip is c. 2000 mm long with a diameter of 50 mm and a total mass of 10 kg.

### 2.1.1 Heating Block

The heating block is designed to dissipate the highest possible heat flux within the smallest possible cross-sectional area. We accommodate six 1 kW heating cartridges in a copper cylinder of 50 mm diameter. The 130 mm-long and 10-mm diameter heating cartridges are custom designed by Freek GmbH (Menden, Germany) to focus heat at the tips of the cartridges. They include a 50 mm unheated area to allow the cable exit points to be distanced safely from the heat generated at the tip. The



heating cartridges are placed as deep into the copper block, or as close to the tip of the melt tip as possible (Figure 1). The six
heating cartridges form three 230 V circuits within a wye/star wired 3-phase plus neutral line configuration.

Following previous melt tip designs, we use a 60º cone to form the bottom of the melt-tip head, to dissipate heat downwards
most effectively into the ice [Talalay et al., 2019], with a parabolic shape forming the rest of the melt tip, which has worked
well in previous melt tips [Kasser, 1960; Hooke, 1976]. In an ideal case, the entire heat source would fit in the 60º cone area.
We must make the copper melt tip long enough to account for the 80 mm heated length of the cartridges to avoid otherwise
overheating the insulated interior of the probe. A small maximum diameter of the melt tip was prioritized. Six flat areas are
removed around the widest point of the copper block, to allow easy grip in a bench vice (two points) or a lathe (three points)
during finishing, final tightening, and sealing.

While the design of the melt tip evolved over four versions during the three-year project life, the fundamental heating block
design of six 1 kW cartridges in a copper block of 50 mm diameter with a 60º cone remained consistent across all four versions
(v0-v3; Figure 2). The v0 was a proof-of-concept that simply allowed all six 1 kW cartridges to be powered up from a variable
power supply, but otherwise contained no electronics package. The v1 tested a PTFE collar around the upper portion of the
copper heating block to better direct heat flow downwards. The v2 discontinued the use of this PTFE collar, increased the
weight considerably and improved the sealing between the copper tip and the steel body. The final v3 increased the exposed
copper heating block area outside of the steel body. While our v3 form is a big improvement over our v1 form, we acknowledge
that both theory and practice suggest that there is still substantial room for improvement of this copper heating block design
[Shreve, 1962; Heinen et al., 2020].

Heat transfer from metal to ice is roughly three times as intensive as from metal to water [Kasser, 1960], so if sufficient
force causes close ice-metal contact at the tip, the highest heat flux should occur within the 60° cone area. The remaining
section of the copper, likely surrounded by meltwater, should serve to build up heat that is directed towards heat loss at the tip,
as well as increasing the total amount of heat delivered into the borehole. This may result in a decrease in total efficiency, but
still will likely deliver faster penetration than otherwise limiting the amount of power we can provide the tip out of fear of
overheating.

### 2.1.2 Electronics

The melt tip houses an electronics package that uses a custom electronics board to monitor temperatures at nine locations in
the probe: four internal temperatures of the heating units, two temperatures at the top surface of the melt tip, and three locally
on the circuit board. The package also tracks the acceleration and orientation recorded by a gyroscope/accelerometer (Figures
3 and 4). We use a BNO080 triaxial accelerometer and gyroscope, which also includes a magnetometer. We install three
ADS1118 ultrasmall, quad-channel, 16-bit analog-to-digital converters to read the differential thermocouples of the four of the
six cartridge heaters plus the two independent thermocouples embedded into the copper heating block. Local temperatures, or
thermocouple "cold junctions", are measured inside the ADS1118 chips. A 12-connection screw terminal block provides a



tight connection for the thermocouple leads. The thermocouple wires are shielded by metal braid, which is electrically connected to the shielding of the data cable, providing protection from electrical noise.

The electronics board continuously reads these six thermocouples and triaxial accelerometer/gyroscope and magnetometer values, transmitting a comma-separated serial stream over RS485 approximately every two seconds. Serial to RS485
conversion is done with an external diode for ESD protection and 120 Ω termination resistance. Twisted pairs from the data cable are soldered directly to the top of the PCB to connect differential RS485 lines and to supply power to the PCB. For first-order noise filtering, 50 Ω resistors and 100 nF capacitors are placed on every connection to data lines, as well as a 1M pullup on CS pins to avoid floating during device reset. These also double as short circuit protection. Testing on early melt tip versions indicated that all data cables, especially those linking the thermocouples with the electronics board, needed to be electrically
shielded to ensure data quality by reducing electromagnetic interference associated with the cartridge heater power cables. At the top-side interface, the serial stream from the melt tip is merged with the serial stream from the winch, to provide a single real-time data stream for operator feedback.

### 2.1.2 Structure

The melt tip has two main sections, each housed within a separate steel body (Figure 5). The lower section, which contains the
copper heating block and electronics package, also contains an additional 2 kg copper weight block. Except for a small air cavity into which the unheated ends of the cartridge heaters protrude, the interior of the lower section is filled with a low-conductivity and high-temperature silicone. An M10 threaded rod running through the probe centre from the copper block to a steel top cap is used to fasten the bottom and top components under high compression. A ~24 mm stack of Belleville washers compressed during top cap tightening accommodates up to 5 mm of expansion associated with thermal and pressure changes.
The upper section lengthens the melt tip to permit vertical stabilization via pendulum steering under gravity [Aamot, 1970; Grzés, 1980]. While pendulum steering is a traditional and reliable approach, it clearly offers no ability to steer the drill against gravity [Dachwald et al., 2014]. The upper member is an unsealed and unfilled steel jacket through which the cables of the lower member pass. With five cables exiting the lower member that require waterproof connections – four cartridge heater power cables and one data cable – there is very limited free cross-sectional area around the central M10 bolt (Figure 6). The
upper section also extends the central M10 bolt to an eyebolt that serves as the structural connection to the winch cable. Both the lower and upper sections have 3 mm thick steel jackets of 50 mm diameter. Together, these members give the melt tip a total length of ~1700 mm, with the eyebolt extending a further ~250 mm.

### 2.2 Umbilical

The umbilical cable serves four distinct functions: powering the cartridge heaters in the melt tip, transmitting signals from the
sensors in the melt tip, providing structural support to control the rate of descent during drilling, and finally deploying the thermistor string that will measure ice temperatures after drilling. Consequently, the umbilical cable consists of four distinct components: power cable, data cable, thermistor cable and structural wire. Together, the four components of the umbilical


cable have an estimated mass of 0.762 kg/m, ignoring the mass of zip ties (Table 1). The idealized cross-sectional area of the umbilical cable – meaning the sum of the cross-sectional areas of the four components – is 276 mm$^2$. The effective cross-

sectional area – acknowledging imperfect fit with ~15% misfit gaps between the four umbilical components – is likely closer to ~386 mm$^2$. The umbilical cable therefore occupies ~20 % of the cross-sectional area of the melt tip (~1963 mm$^2$).

The power, data and thermistor cables are arranged at the ice surface to separately feed each into the borehole. All three cables are zip-tied to the main structural wire, every ~50 cm during drilling, as it passes through the winch frame (Figure 7). Given the appreciable resistive heating and electromagnetism generated in the power cable, the power cable must be unspooled

when conducting ~6 kW power, to prevent melting the cable insulation and producing a large coil effect (Figure 8). Ideally, one should unspool the power cable and arrange it in an ~3 m 'figure eight' formation on the ice outside the drill tent, as a 'figure eight' formation should cancel induced electric fields where the power cable crosses itself. In practice, however, it is easier to just have large loops of power cable on the ice surface. As the data cable and unpowered thermistor cables experience negligible resistance heating during drilling, they can be deployed from spools.

### 2.2.1 Power Cable

The power cable, which powers the cartridge heaters, is the largest and heaviest component of the umbilical. The primary design requirement of the power cable is four conductors capable of powering the wye/star wiring of three sets of 1 kW cartridge heater pairs. The secondary design requirement is the rather extreme range of operating temperatures that may be encountered. We use the ÖLFLEX Robust 210, which comprises four individually shielded fine-wire copper conductors, each

with a cross-sectional area of 10 mm$^2$. This product has a TPE outer sheath that is environmentally and chemically resistant. The TPE sheathing is rated to maintain its flexibility and integrity to a lower operating temperature of -40 °C.

The top-side connection of the power cable is a conventional 400 V three-phase plug-and-socket connection to the interface. The bottom-side connection of the power cable is four individually crimped connections to the four power cables within the upper member of the melt tip. Each crimped connection is sealed with heat shrink tubing that contains adhesive glue. To

minimize umbilical diameter, the four crimped connections are distributed over ~1 m of power cable. This ensures that the relatively thick connections are non-overlapping and can fit within the upper member of the melt tip. While there are certainly more sophisticated and elegant methods to achieve water-resistant electrical couplings, such as pressure-rated male-female couplings, crimps and heat shrink tubes provide a relatively simple bottom-side connection that can allow melt tips to be attached under field conditions. The relevant pressure-rated connectors also have a non-trivial diameter (>25 mm) that presents

a challenge to both the available cross-sectional melt tip area and the borehole diameter.

There are no external transformers used in the power system. Conducting electricity over relatively long distances results in non-trivial power and voltage loss due to resistance. Technical specifications suggest 6.5% voltage losses over a 500 m umbilical length at 100% power. This results in a net resistive power loss of 386 W per 500 m and decreases the effective voltage from 230 V to 215 V at the cartridge heaters. The power cable has an upper operating temperature of 80 °C, which

means it can withstand limited resistive heating. When in air, 500 m of unspooled power cable in 'figure eight' formation


would therefore technically require a passive environmental heat sink of at least 386 W. Resistive overheating of the power cable heating is not an issue when it is submerged in the water-filled borehole.

## 2.2.2 Data Cable

The data cable returns signals from the instrument package in the melt tip to the interface at the surface. The primary design
requirement of the data cable was transmitting data signals in low voltage RS485 protocol without interference from the induced electromagnetic field of the adjacent high voltage power cable. These data signals include real time-measurements of the orientation data string and the thermocouples embedded in the cartridge heaters. For the data cable, we use an Etherline Robust PN Cat. 7 cable that has four shielded pairs of copper wire. In addition to physical shielding of the data signal, the RS485 transmission protocol makes the data signal relatively insensitive to low-frequency noise. Any 50 Hz interference from
the AC power cable should influence paired RS485 signal wires in a compensating fashion.

The top-side connection of the data cable is a standard male-female connector to the interface. The bottom-side connection of the data cable is an IP67 rated male-female RS485 connection. The female end of this connection is embedded in epoxy and emerges at the top of the melt tip. The male end is attached to the data cable. These multi-pin connections make the data cable connection the most complex connections of the four umbilical components. It is therefore challenging to modify data
cable length under field conditions; altering data cable connections generally requires workshop conditions. The outer sheath of the data cable is PTE-V that is rated to maintain flexibility to a lower operating temperature of -40 °C. While the data cable is only used to transmit signal during drilling, it can potentially be reactivated to measure the embedded melt-tip thermocouples any time after drilling. In this way, the data cable provides a secondary mechanism for measuring ice temperature that is completely redundant to the thermistor cable.

## 2.2.3 Thermistor Cable

The thermistor cable, which is deployed with the umbilical cord during drilling but only starts to measure and log ice temperatures once drilling has finished, represents the main sensor suite of the drill system. The primary design requirement for the thermistor cable is that, with ultra-low power consumption, it can measure and log precise ice temperatures for the several months required for the thermal disturbance of drilling to slowly dissipate to background ice temperatures. The
thermistor cable does not enter the melt-tip. Therefore, it has no bottom-side connections, it is simply structurally attached to the umbilical with zip ties. After insertion into the ice, the topside of the thermistor cable can be connected to a solar-powered automatic data logger [Fausto et al., 2021].

The thermistor cable consists of a multi-lead wire with attached thermistors. We use an Alpha Wire multi-lead cable that has 12 individually insulated copper wires. This allows attaching 11 thermistors at desired intervals along the thermistor cable.
To ensure a closer spacing of thermistors near the ice bed, where vertical gradients in ice temperature are anticipated to be greatest, the optimal spacing of these thermistors should form an exponential decay with ice depth. K-type thermistors are manually spliced into the desired positions of the multi-lead cable and then wrapped in heat shrink tubing. The outer sheath of





the thermistor cable is PVC, which gets inflexible and brittle below -20 °C. While this type of thermistor cable has been used extensively by GEUS for ice-sheet monitoring stations, digital temperature sensors would allow many more temperature

measurements to be collected using a multi-lead cable with even fewer wires [Li et al., 2021a].

To recover precise ice temperatures, the thermistors are rapidly equilibrating (10 s) and the thermistor cable has no shielding that could vertically conduct heat. This lack of shielding, however, means that the thermistor cable does not return reliable data during drilling, when the adjacent power cable induces a strong electromagnetic field. The thermistors have a precision of ±1 %, or ~0.01 °C, over an operating range of -80 to 150 °C. To maintain ultra-low power consumption, the

thermistor cable operates on just 2.5 V. As thermistor resistance varies as a function of ice temperature, the data signal can be very sensitive to resistive loss in the signal wire. Using a relatively large cross-sectional area (0.35 mm$^2$) for the relatively low voltage (5 V) in the signal wire minimizes resistive voltage loss to <<1 % over 500 m.

### 2.2.4 Structural Wire

The structural wire bears the combined weight of the melt tip and umbilical. The primary design requirement of this wire is

sufficient strength with minimum diameter. We use a steel aircraft cable that has seven bundles of nineteen galvanized wires. This aircraft cable is supplied by US Cargo Control as 3/16" wire, equivalent to a metric diameter of 4.76 mm, and has an ultimate tensile strength of 18.6 kN. The bottom-side termination of the structural wire is a loop that passes through the eyebolt protruding from the melt-tip. The loop is held by brass wire clips, which permit free movement around the eyebolt. The top-side connection of the structural wire is a secure clamp on the winch drum. Once drilling has started, the length of the structural

wire provides a fundamental depth limit for the drilling system.

The structural wire bears 0.448 kg/m of all four umbilical components (Table 1). It would clearly be desirable to have an integrated umbilical that combines these four components into a single cable [Peng et al., 2021; Zhang et al., 2021]; as the combined mass of the power, data and thermistor cables is within the structural tolerance of some off-the-shelf power cables. As previously described, however, resistive heating prevents a power cable from being tightly spooled on a winch drum, at

least without an active cooling system. As the structural wire is the only umbilical component to pass through the winch, it controls the descent rate of the melt tip and umbilical. This control function is arguably a more unique contribution to the drilling system than its structural function.

### 2.3 Winch

The main function of the winch is to control the descent rate of the melt-tip and umbilical (Figure 9). It serves as the physical

connection between the interface and the umbilical. The primary design requirement of the winch is ensuring accurate pay out of the structural wire under sufficiently high torque. A secondary design requirement is minimizing the weight and volume of the winch, with an upper length limit of 130 cm to facilitate fieldwork transportation. Finally, the winch also provides open access on two sides above the borehole, to allow the other components of the umbilical (power, data and thermistor cables) to





be manually zip-tied to the winch's structural wire. With an empty spool, the winch weighs a total of 80 kg, consisting of 65
kg of frame and motor and 15 kg of timber foundation. It runs on 230 V and draws a maximum of 500 W.

### 2.3.1 Frame

The main design requirement of the frame is bearing the load of the umbilical. We considered using an adjustable T-slot
modular frame with T-nut fasteners but found that a welded steel tube structure was easier to render in the structural analysis
software needed to understand how the wire load produces vertical stress in some elements of the frame and horizontal stress
in other elements of the frame. The winch frame is therefore constructed with square tube steel with a thickness of 1.65 mm
and welded joints. We estimate the safety factor of a welded steel frame as ~3 when the winch is under 400 kg load. The steel
frame is secured to a timber foundation of three 100 x 100 x 2000 mm cross members. This timber foundation increases
stability and decreases ground pressure. Under characteristic load of a 500 m borehole, these timbers would ultimately bear up
to 315 kg (65 kg winch weight, 10 kg melt tip weight, 15 kg foundation weight and up to 225 kg umbilical weight) and exert
a pressure of 17 kPA (2.5 psi) on the underlying snow or ice.

The length of the winch is determined by the 130 cm design requirement, and the desire to provide the greatest distance
possible between the encoder pulley and the winch spool. Maximizing this distance, along with using a relatively narrow but
deep spool, minimizes variations in the feed angle of wire from the spool to the encoder pulley. The winch spans the borehole
along its length axis, as the load is too great to cantilever over from one side of the borehole. The 800 mm width of the winch
is determined by the need for a horizontal outrigger to accommodate both the spool-perpendicular motor, as well as possible
off-axis side-loading when working on snow and ice surfaces. This outrigger ensures that the winch should operate with a
cross-slope tolerance of up to ~3°. Due to space and weight considerations, the height of the winch is minimized, at 560 mm.

### 2.3.2 Motor

The main design requirement of the motor is ensuring sufficiently high torque while still ensuring sufficiently small increments
of pay out from the winch spool. This requirement ultimately determined a single gear system, in which a stepper motor was
paired with a 120:1 gear reduction. This ensures that each of the integer steps taken by the motor can be further reduced into
a mm-scale unit of pay out. As the spool radius changes through time because wire slowly pays out from the spool, the wire
passes through an encoder pulley to independently measure pay out length. During normal operation, there is no brake; the
motor and gearbox together provide sufficient high-torque position control. The gearbox is the least cold-tolerant winch
element, with a rated lower operating temperature of -25 °C. The ambient temperature of the drill tent is usually above this
threshold.

The single gear system can lower or raise the wire at prescribed speeds of up to 30 m/hr, within an estimated precision
of 0.01 m/hr within the prescribed speed. The winch does not have a second, higher, gear for spooling the cable. At maximum
speed, the motor has a maximum pulling force of 5.8 kN, when all the cable is fully spooled, and 11.3 kN when the cable is
fully deployed. As the drilling system is intended for the one-way deployment of melt-tips and thermistors into the ice, the



winch also does not have an auto-leveller when pulling in wire to the spool. The lack of an auto-leveller permits the winch spool to be relatively deep and narrow, reducing the set-back distance between the spool and encoder pulley. With a high gear and auto-leveller, however, the winch must be decoupled from the motor hand spooled. We estimate that removing the winch drum from the drivetrain, using a removable shaft collar, and spooling a new wire takes a non-trivial three person hours.

### 2.2.3 Electronics

The winch electronics provide the interface control over, and feedback from, the winch. They are housed in a small Pelican case attached to the frame that communicates with the interface via plaintext RS232 protocol. This communication allows the interface to prescribe winch speed (within ±30 m/hr) and log the pay out from the encoder pulley and load from the load-pin pulley. The accuracy of the load pin is doubled by looping the wire 180 ° around the load-pin pulley. The load-pin pulley measures absolute load to within ±6 kg accuracy and relative load changes with <1 kg accuracy. The encoder pulley measures pay out to better than ±1 mm accuracy. Until field conditions, however, the metal-on-metal combination of the encoder pulley and the structural wire allowed slippage, which substantially reduced the reliability of the digitally recorded pay out. Manual logging ultimately proved to be the most reliable depth estimate.

The winch motor and electronics are powered by a single 220 V cable from the common power supply. The winch motor draws a maximum of 500 W. The winch electronics draw negligible power. There are, however, separate ground lines for the motor and electronics. This protects the electronics package from any electrical noise and surges generated by the motor. In the event of an emergency, loss of power, or electrical fault, a power-cut button is depressed to engage an electromagnetic emergency brake on the motor. While there is no manual control over the winch – all instructions must be digitally transmitted via the interface – the open-source Arduino software used by the winch electronics is easy to modify and upload under fieldwork conditions. The power-cut button proved to be the easiest way to stop and restart descent during ice-sheet drilling.

### 2.4 Interface

The primary design requirement of the interface is to manage both high-voltage power and low-voltage data signals from all drill system components in a compact and operator-friendly fashion. The interface consists of three distinct components: melt-tip power unit, melt-tip data unit, and winch control unit (Figure 10). A field laptop can be connected to the interface to log data and provide digital commands. In the absence of a field laptop, the drilling system can be operated with manual controls and no data logging. Including a laptop, the interface weighs approximately 20 kg and uses less than 100 W of power. All interface components have a lower temperature limit of at least -20°C.

While the interfaces are watertight when closed during shipping and transport, they must be open during field operation to allow the operator to see three independent local real-time data displays and accommodate many diverse connections between the interface and other drill system components. These connections include power in from the generator, power out to the umbilical cable, data from the umbilical cable, data out from the power unit, data in/out with the winch, two USB connections





to a field laptop, and its own 230 V power supply. Clearly, it would be desirable to migrate the eight connections into a series of glands and ports on the exterior of a waterproof housing to make the interface more robust and water resistant (Figure 11).

### 2.4.1 Melt-Tip Power Unit

The melt-tip power unit is the regulator between the raw generator power output and variable power delivered to the umbilical cable. The primary purpose of this unit is to provide operator control over power delivery within the 0 to 6 kW range, while also protecting the generator by evenly distributing load and minimizing demand spikes. The heart of the melt-tip power unit is three 3-phase silicon-controlled relays (Thyro-A 3A 400-8 HRL3) which efficiently adjust the power supplied to the umbilical cable. The three 400 V line voltages from the generator are wired in a wye/star configuration, supplying 230 V across

the heating elements with a common neutral line. Instead of using internal melt-tip temperature as a thermostat, regulating the generator power on/off, the relays analyse the generator's 50 Hz AC signal and deliver a percentage of each cycle as set by the operator.

The drill operator can prescribe power delivery as a percentage within the 0 to 6 kW range via a field laptop connected to the interface. The power unit logs power delivery along with all settings and modes of operation through time. This data is

sent to the field laptop via USB connection using Thyro-Tool software. A key safety feature of the power unit is an independent circuit breaker (Lovato SM1R 2300) on each of the three incoming 230 V lines from the generator. These circuit breakers exist mainly to protect the operators and generator from short circuits. The silicon-controlled relays have their own circuit breakers that protect the interface in the event of a down borehole short circuit from water ingress in either the umbilical cable or melt tip. Fuses are fitted in each relay in the event of circuit breaker failure or generator surges.

### 305     2.4.2 Melt-Tip Data Unit

The primary task of the melt-tip data unit is receiving serial data from the umbilical data cable that transmits these data from temperature and orientation sensors embedded in the melt tip (Figure 12). To receive this data, the melt-tip data unit must first supply the umbilical data cable with regulated 12 V to power the electronics package within the probe. This interface contains an RS485-to-USB converter, which allows the attached field laptop to monitor and log the received information using

SerialStudio software. The melt-tip power unit is the regulator between the raw generator power output and variable power delivered to the umbilical cable.

### 2.4.3 Winch Control Unit

The primary task of the winch control unit is to provide reconfigurable messaging to the winch, as well as receive real-time feedback from the winch. The winch has its own firmware to convert its step-motor rotation into pay out length as a function of spool diameter. The winch control unit simply instructs this firmware of the desired speed and direction of pay out. This

instruction is transmitted via RS232 messages sent from the interface using an Arduino Mega. The winch control unit also



locally displays the incoming and outgoing data stream on a small colour TFT display. This data stream includes the pay-out distance, winch speed, winch direction, load cell values and deviation of load cell.

In the absence of a field laptop, the winch control unit of the interface has glove-friendly manual controls to select winch speed and direction. The winch control unit then converts these manually selected values into digital messages and sends them to the winch. The RS232 connection between the interface and the winch is isolated to avoid creating a ground loop. When resetting the power supply, during generator refuelling for example, all interface data streams reset. This creates the largest issue for the winch data stream, as every power reset ultimately resets the winch pay out/depth back to zero.

**2.5 Power Supply**

The power system powers the melt tip, the winch, and the interface during drilling. The theoretical maximum power requirement of the drilling system, including resistive loss in the umbilical and peripheral devices, is approximately 6.6 kW (Table 2). The primary design requirement of the power system is therefore supplying sufficient quantities of both 3-phase 400 V and 2-phase 230 V power. While multiple smaller generators in parallel could meet this requirement, a parallel generator configuration requires a synchronometer, which represents an additional potential failure point [Anker et al., 2021]. We

therefore use a single large generator. The Pramac S12000 has a rated maximum output of 11.1 kW and a rated continuous output of 9.5 kW. This continuous output provides 2.9 kW more power than the theoretical maximum 6.6 kW required by the drilling system; this represents a capacity excess of ~44% over a 6.6 kW demand baseline.

The generator is the single heaviest and largest component of the drilling system; it weighs 165 kg and has dimensions of 960 x 641 x 667 mm. Despite its size, however, the generator can still be loaded into a DHC-6 Twin Otter aircraft with an

effort level similar to loading a snowmobile. As the gasoline (or petrol) variant of the Pramac S12000 is slightly lighter than the diesel variant, we selected the gasoline variant. The generator clearly requires an appreciable fuel supply to sustain kW-scale output for long periods of time. We measure a fuel consumption rate of 4.5 L/hr for a sustained 5.5 kW output, which is slightly less than the 6.6 kW theoretical maximum of the drilling system. An ~50 hr drilling project therefore requires ~225 L of fuel. With a tank capacity of 24 L, this requires re-filling the generator every ~5 hr. Ongoing refuelling is hazardous, as the

generator's fuel tank is located immediately above an exhaust pipe that has an operating temperature of > 230 °C (Figure 13). This is approaching the auto-ignition temperature of wayward gasoline droplets (280 °C). It is important to completely fill the generator's fuel tank at the end of each day. This expels as much air as possible from the tank's headspace, to prevent the development of ice in the tank, which can block the fuel line. We found that the generator was difficult to start when the ambient air temperature was below -10 °C.

Due to concerns over both noise and exhaust pollution, the generator is housed in a separate generator tent ~30 m downwind from the drill tent. The generator has a rated noise level of 68 dB at 7 m. Moving the generator to a distance ~30 m away theoretically reduces this noise level to 54 dB. The tent wall of the drill tent, as well as an intervening snow wall, further reduce this noise level to an estimated 40 dB. To ground the generator, we drill a 6 m aluminium pole into the ice sheet and pour water into the borehole to refreeze and increase local conductivity around the base of the pole. We then attach the ground or earth



line of the generator to this pole. In firn-covered areas of the ice sheet, substantial water volumes may be needed to increase the local near-surface conductivity of relatively high porosity firn to provide sufficient grounding for the generator.

## 2.6 Support Items

In addition to the five major components described above, deployment of the melt-tip drilling system requires several minor support items. A Mountain Hardwear Stronghold tent, with a peak interior height of 196 cm and an interior floor area of 15.9

$m^2$, serves as the drill tent. A smaller Pop'n'Work (GS8612) tent, with 4.4 $m^2$ floor area, serves as the generator tent when needed due to weather conditions. While the drill tent is double-walled and can maintain a comfortable working temperature, the downwind wall of the generator tent is typically open for ventilation during operation. Neither tent has an in-built floor. To comfortably place the generator tent ~30 m downwind from the drill tent, we use heavy duty 50 m extension cables for both the 400 V and 230 V power supplies. We use bamboo poles of ~1.5 m length, paired to form X-shaped holders, to keep

the power lines elevated above the snow surface and avoid their freeze-in.

While we bring a wide range of hand tools for servicing the various components of the drilling system. One especially invaluable tool is a thermal camera (Seek Thermal ShotPRO). The thermal camera can quickly, and independently, check the temperature of components that have high operating temperatures, including the power cable, interface transformer and generator. Unfortunately, it proved nearly impossible to accurately measure the temperature of the copper heating block

protruding from the melt tip, presumably due to complex emissivity properties of the wet metal surface.

In all, these support items weigh ~150 kg, excluding fuel. Rather than transporting fuel in 200 L drums, we prefer to use 20 L metal canisters, each of which have a tare weight of 4.3 kg. Accounting for 65 kg of metal canisters means that a generous fuel budget of 300 L weighs 290 kg total. In total, all drilling system components required for 500 m of drilling weigh ~1090 kg (Table 3). The drilling system is therefore within both the mass and volume limits of a single ski-equipped DHC-6 Twin

Otter flight. This total system weight is comparable to other lightweight hot-point drilling systems of similar depth capability [Zagorodnov et al., 2014].

## 3 Laboratory Testing

Laboratory testing of the v2 melt-tip was performed in the ice well of Jilin University, China, in September 2021. The artificial ice well is 12 m deep and 1 m wide, with an ice temperature of approximately -10 °C [Wang et al., 2018]. During testing, the

melt tip was fastened to 2 m metal drill flight suspended from a winch above the ice well. An encoder on the winch provided an independent estimate of penetration rate during a ~2 m penetration test (Figure 14). Two tests each were conducted at power levels of 1.1 kW (18%), 1.9 kW (32%) and 2.7 kW (45%). More tests were intended at higher power levels, but the melt tip failed with an electrical short during a 4.5 kW (75%) power test. During these tests, penetration rate was measured every 1 second with 1 mm accuracy. The resulting mean penetration rates range from 1.9 m/hr at 18% power to 5.9 m/hr at 45% power.

In comparison to the theoretical maximum penetration rate associated with perfect heat transfer efficiency between the melt tip and the surrounding ice, these laboratory penetration rates suggest the melt tip has a ~35% heat transfer efficiency (Figure





15). Assuming that penetration rate is linearly proportional with power [Li et al., 2021a], the maximum penetration rate under laboratory settings would be ~12 m/hr.

The cause of the melt tip's electrical failure during the 75% power test was explored via destructive analysis of the v2 melt tip. This analysis suggested that the internal resin fill, which was in contact with the top of the copper heating block, had melted. This caused a mechanical instability, whereby the copper heating block could move within the steel jacket, which allowed water to enter the melt tip at the copper-steel seam a short the cartridge heaters. The resin fill was not similarly compromised higher in the melt tip (Figure 16). Subsequently, a different type of silicone fill was chosen, and the copper heating block of the v3 melt tip was extended upwards, further into the steel jacket, to reduce the unheated portion of the cartridge heaters exposed from the top of the copper heating block. A small air gap was also introduced between the top of the copper block and the resin fill. Finally, hexagonal mounts were added to the exterior of the copper heating block to enable a higher-pressure seal when tightening the top cap against the M10 bolt mounted in the copper heating block. These design changes stemming from laboratory failure were likely critical in ensuring that the v3 melt tips did not suffer similar electrical failures during field testing.

## 4 Field Testing

Initial integrated testing of the ice-drilling system with a v3 melt tip was performed at Tuto Ramp, near Thule Air Base, Greenland, in May 2022 with two operators. This testing brought together the melt tip, umbilical cable, winch, interface, power supply and support items described above for the first time. The melt-tip drill was over shallow depths (< 1 m) at variable cartridge heater power levels in lake ice. The goal of these tests was to understand the borehole width as a function of cartridge heater power level. Tests were performed at 1.5 kW (25%), 3.0 kW (50%) and 4.5 kW (75%) power. While the rate of penetration was clearly proportional with power level (3.0, 4.4 and 5.6 m/hr, respectively), the borehole diameter was a remarkably constant ~70 mm diameter over all tests (Figure 17). We interpret this to suggest that the 50 mm diameter melt tip has an effective borehole diameter of 70 mm. These field tests on lake ice suggest that the melt tip has a ~25 % heat transfer efficiency.

The ice drill was more extensively tested at an ice-sheet ablation area site known as D-11 (76.4109°N and 68.2876 °W), where an ice-sheet borehole and temperature profile had been previously measured by the US Army Corps of Engineers (USACE) in 1961 (Figure 18). In 1961, the USACE estimated the ice thickness at D-11 to be 52 m. Based on the 2022 elevation difference between the ice surface at D-11 and the adjacent ramp road surface, we expected the 2022 ice thickness at D-11 to be ~25 m, which would be consistent with ~27 m of ice surface melt since 1961. Curiously, however, our 100 MHz ice-penetrating radar system suggested a 2022 ice thickness at D-11 of ~44 m. The radar data, however, suggests there is a possible reflector horizon about ~20 m above the true bedrock at D-11 (Figure 19). It is therefore conceivable that the mechanical drill of the USACE struck 'bottom' in an englacial debris layer, rather than the true ice-sheet bed.

For ice-sheet drilling at D-11, the winch foundation was located on the snow surface. A mechanical drill of Ø 50 mm was used to drill through the ~1.5 m thick winter snowpack, to provide the melt-tip drill with direct contact to the underlying


glacier ice. During drilling, the power supply of the melt-tip was slowly increased, in ~10-minute intervals, to find a practical equilibrium between winch pay out and downward load on the melt tip over increasing penetration rates. For this initial testing, thermistor cables were not employed, so the umbilical bundle consisted of only one power cable, one data cable and the main structural wire. The main goal of the ice sheet field testing was to observe the performance of the drilling system for longer durations and higher power levels than laboratory testing.

Two boreholes were drilled at D-11. The shallower borehole only reached 5 m depth with a mean rate of penetration of 1.7 m/hr and mean power of 3.0 kW (50%). The cartridge heater temperature varied between 200 and 300 °C for ~3 hours. Despite stable melt-tip temperatures and a stable rate of change on the cable load, the rate of penetration was remarkably slow throughout the drilling of this borehole. After 5 m, drilling was stopped, and a camera was lowered into the borehole (Figure 20). The camera revealed that ~1 cm of sediment had already collected on the borehole bottom, which greatly reduced

downward heat transfer and penetration rate [Li et al., 2020]. A water-filled cavern of diameter ~50 cm had formed around the melt tip during this time. We suspect the borehole was initiated in a sediment-filled topographic low point on the snow-covered ice-sheet surface. We then realized that drilling between two ramp roads constructed by the USACE hauling thousands of tonnes of road fill onto the ice sheet was poor site-selection for a melt-tip drilling system that was sensitive to borehole sediment. Generally, however, this borehole confirmed that the melt tip could sustain multi-hour operations without

overheating.

The deeper D-11 borehole, which was located just 2 m from the shallower borehole, reached 21 m depth with a mean rate of penetration of 2.3 m/hr and mean power of 4.2 kW (70%). The melt-tip cartridges again sustained operating temperatures of between 200 and 300 °C for ~9 hours while drilling this borehole. The rate of penetration was consistent until an abrupt stop at a borehole depth roughly consistent with the layer that the USACE reported as the ice bed. Neither leaving the melt-tip

in contact with this layer for an hour, nor manually raising and dropping the melt-tip, could penetrate this layer. From this deeper borehole, we can also estimate a peak 1-hour sustained rate of penetration of 4.5 m/hr at 5.1 kW (85%) power. Rates of melt tip penetration observed in laboratory conditions are generally not directly comparable to those observed under field conditions [Gillet et al., 1984]. For example, when drilling over greater depths in the field, increasing inclination angle was countered by raising the melt tip c. 1 m and then lowering it again at normal penetration rate to correct inclination angle. While

this approach was successful in returning the borehole to plumb, it means rates of penetration in the field are only a fraction of penetration in a laboratory. The 21 m borehole, for example, suggests that the melt tip has a ~15 % heat transfer efficiency. By comparison, some melt tip drills can achieve >80% heat transfer efficiency under field conditions [Hooke, 1976].

## 5 Discussion: Borehole Refreezing

Borehole refreezing is the fundamental depth limitation of melt-tip drilling when the umbilical pays out from the ice surface

[Zagorodnov et al., 2014]. Surface deployed umbilical cords are continually moving downward relative to the surrounding ice. This requires an unfrozen borehole between the ice surface and the melt tip to allow continued descent. Philberth-type melt tips accommodate borehole freezing by unspooling their umbilical cable from within the melt tip. In these melt tips, the





umbilical cable does not move after it leaves the melt tip and is therefore unaffected by refreezing of the borehole [Aamot, 1967]. As the system that we describe here has no mechanism to counter borehole refreezing, such as anti-freezing or electrical

heating, its maximum theoretical penetration depth depends on outracing the refreezing front within the uppermost, or oldest, portion of the borehole [Suto et al., 2008; Hills et al., 2020].

We simulate borehole refreezing using a 1-D radial heat transfer model. This model is relatively simple in comparison to other models describing processes at the interface between the melt tip and surrounding ice [Li et al., 2021b]. Our model consists of an enthalpy-based formulation of heat diffusion and latent energy exchange across a moving water-ice phase

boundary [Greenler et al., 2014]. This formulation ignores heat advection, assuming the borehole water is essentially stagnant, as well as heat production with the umbilical, as frictional heating and resistive loss are comparatively negligible. The radially symmetric two-phase moving boundary solution is solved at 1 mm radial spacing and 100 second time-steps with an implicit formulation (backward Euler). We prescribe the initial borehole water temperature at 75±25 °C and initial borehole radius as the 25 mm melt-tip radius. We explore the change in borehole radius over ice temperatures between -1 and -20 °C.

Under all ice temperatures, the borehole radius initially grows rapidly, due to the high temperature gradient across the water-to-ice interface. After this brief borehole growth period, during which heat is rapidly diffusing into the ice, a longer borehole decay period begins. As the borehole refreezes, the radius decays at an increasing rate, until closure when the radius reaches the 10 mm radius equivalent to the effective cross-sectional area of the umbilical. Due to the high specific heat capacity of ice, the maximum borehole radius and decay time to 10 mm radius are both highly dependent on ice temperature (Figure 21). In -

5 °C ice, the maximum borehole radius reaches 46±4 mm and the borehole decay time is 17±4 hours. In -15 °C ice, these values are only 35±3 mm and 4±2 hours, respectively.

Over the ice temperature range of -1 to -20 °C, the borehole decay time to a 10 mm radius decreases from ~20 hours to ~4 hours. In temperate ice, at the pressure-melting-point of 0 °C at 1 atmosphere pressure, the borehole would never refreeze. Interestingly, feedbacks between the rate of borehole growth and the ice temperature result in a local minimum in refreezing

time at ice temperatures of between -1.5 and -2.0 °C. Multiplying these refreezing times by a characteristic penetration rate of 10 m/hr (i.e., ~85% of the theoretical maximum 12 m/hr) suggests that the theoretical maximum depth to which we could expect the ice-drilling system to penetrate under ideal conditions without antifreeze is ~200 m. As meltwater can escape the borehole in porous firn, the theoretical maximum depth of the drilling system is the maximum ice thickness plus the porous firn depth (Figure 22). As porous firn depth increases with decreasing near-surface temperature, decreasing theoretical

maximum ice thickness is partially offset by a reasonable assumption of characteristic porous firn depth [Vandecrux et al., 2019].

## 6 Summary Remarks

Here, we have described the design and performance for a new melt-tip ice-drilling system. The system consists of a melt tip, umbilical cable, winch, interface, power supply, and support items. The melt tip and the winch are the most novel elements of

the drilling system. Consistent with an open science mandate, we make the CAD designs for these components available in





the open-access GEUS Dataverse repository: https://doi.org/10.22008/FK2/DXXR06 [Colgan et al., 2022]. We hope that the level of detail that we have provided here is sufficient to allow other groups to leapfrog this drilling technology.

The laboratory ice well testing suggests the melt tip has an electrical energy to forward melting heat transfer efficiency of ~35% with a theoretical maximum penetration rate of ~12 m/hr at 6.0 kW (100% power) under laboratory conditions. The

available literature suggests that there is substantial room for improvement in this heat transfer efficiency (Figure 23). For example, by changing the heating block material and design, or potentially converting to an DC power system [Peng et al., 2021]. The ice-sheet testing suggests the melt tip has an analogous heat transfer efficiency of ~15 % with a theoretical maximum penetration rate of ~6 m/hr. We expect the efficiency gap between laboratory and field performance to decrease with increasing operator experience drilling over greater depths.

In the future, we envision moving towards an integrated umbilical cable and developing an autonomous drilling feedback loop, whereby winch pay out varies as a function of winch load. We are also developing ideas about suitable chemical agents to counter borehole refreezing [Zotikov, 1979; Zagorodnov et al., 1994; Hills et al., 2020]. In the near term, however, we hope to use the ice-drilling system, as described here, to insert 100-m scale thermistor strings into the Greenland ice sheet and/or peripheral ice caps. There are many places in Greenland, far from existing measurements, where a borehole in relatively

thin and slow flowing ice would yield novel insight about ice-bed temperature and geothermal heat flow. Our melt-tip ice-drilling efforts are now transitioning from a development phase, in which the goals have been largely engineering, into an operational phase, in which the goals are largely scientific.

## Acknowledgements

We are exceptionally grateful to the Experiment Programme of the Villum Foundation (award 00022885) for funding the

development of the Hotrod melt-tip ice-drilling system. We thank Jens Bisgaard (GEUS) for metalworking. We thank Dirk van As (Greenland Guidance) and Victor Zagorodnov (Cryosphere Research Solutions) for technical expertise consulting on various aspects of the project. We thank Jack Brandis, Thorbjørn Flegal and Ferando Sevilla (all Copenhagen School of Design and Technology) for insights from a group thesis, completed along with M.J., on melt-tip optimization. Finally, we thank Jørgen Peder Steffensen (University of Copenhagen) and Henrik Højmark Thomsen (GEUS) for collegial discussions on

various aspects of ice drilling.

## Author Contributions

W.C. is lead scientist of the Hotrod drilling system. C.S. is lead engineer of the Hotrod drilling system. A.P. and K.M. primarily contributed to the design of the melt tip, umbilical cord, and power supply. A.L. and J.E. primarily contributed to the design of the winch. H.R. formulated and implemented the two-phase moving boundary heat transfer model. M.J. and M.B. completed

individual theses on various aspects of the project. P.T., X.F., Y.L and X.W. were responsible for laboratory testing in the ice well at Jilin University. W.C. and C.S. were responsible for field testing at Tuto Ramp, Greenland, with support from H.S. N.K. performed radar analysis and co-supervised M.B.



**Data Availability**

All source files for the ice-drilling system, including (1) design files for the winch, (2) machining files for the melt tip, (3)
design files for the internal melt-tip electronics and instruments, (4) software codes for the interface, and (5) numerical codes
for borehole refreezing simulations, are available on the GEUS Dataverse at https://doi.org/10.22008/FK2/DXXR06 [Colgan
et al., 2022].

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





**Table 1 – Dimensions and mass of umbilical components. *Assumes the idealized diameter and cross-sectional area that would be achieved by combining all four umbilical cable components into a single cross-sectional area without misfit gaps.**


| Component | Diameter (mm) | Area (mm²) | Mass (kg/m) | Mass (kg/500m) |
|---|---|---|---|---|
| **Structural Wire** | 4.76 | 17.79 | 0.097 | 49 |
| **Power Cable** | 16.50 | 213.82 | 0.541 | 271 |
| **Data Cable** | 8.70 | 59.45 | 0.076 | 38 |
| **Thermistor Cable** | 7.50 | 44.18 | 0.047 | 24 |
| **Total** | **18.73*** | **335.24*** | **0.762** | **381** |


**Table 2 – Theoretical maximum power demand of the drilling system, as well as generator supply and residual net capacity. Demand denoted as negative and supply denoted as positive.**

| Component | Power (W) |
|---|---|
| **Melt-tip** | -5582 |
| **Umbilical** | -386 |
| **Winch** | -500 |
| **Interface** | -100 |
| **Total Demand** | -6568 |
| **Total Supply** | +9500 |
| **Net** | **+2932** |




**Table 3 – Drilling system weight associated with 500 m of umbilical cable.**

| Component | Mass (kg) |
|---|---|
| **Melt tip** | 10 |
| **Umbilical** | 225 |
| **Winch** | 75 |
| **Interface** | 20 |
| **Generator** | 165 |
| **Fuel** | 290 |
| **Support Items** | 150 |
| **Total** | **1090** |






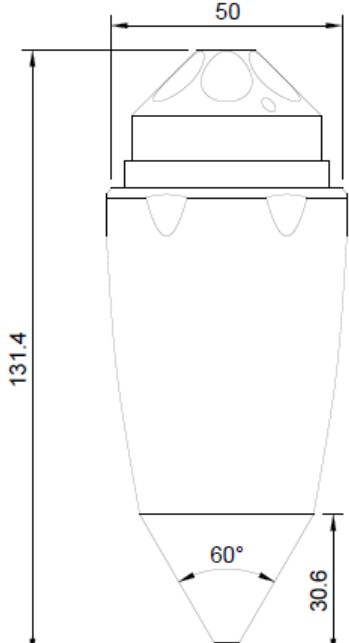

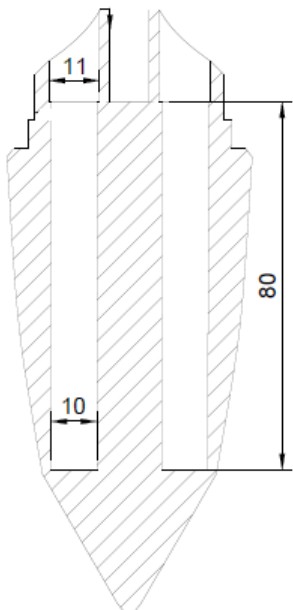

**Figure 1 - Left: Technical drawing of the basic dimensions and sections of the melt tip. Right: Interior view of cartridge slots shows the depth placement of the 80 mm heated section of the cartridges.**



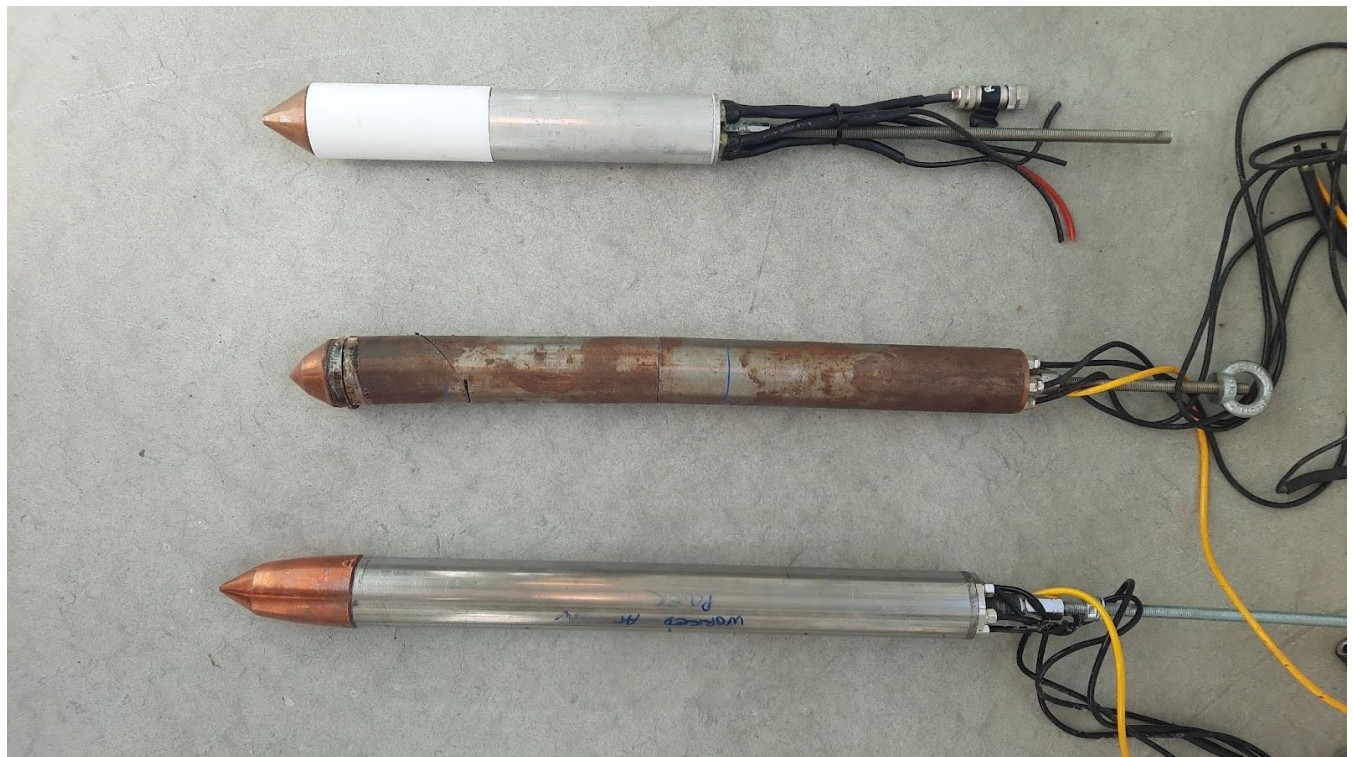


**Figure 2 – Photograph of the v1 (top), v2 (middle) and v3 (bottom) melt tip versions developed over the three-year project lifetime (v0 not shown). The v2 melt tip shown here, which has been sectioned for destructive testing, bears a characteristic rust coating that develops with melt tip use. The v1 and v3 melt tips shown here have not been used.**



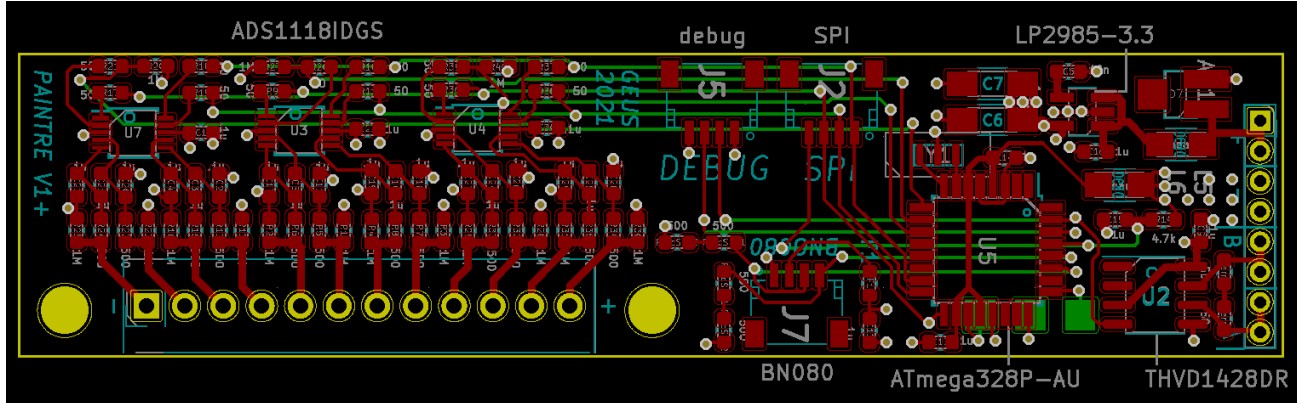


**Figure 3 – Customized electronics board within the probe that integrates measurements from the cartridge and independent thermocouples as well as the triaxial accelerometer and gyroscope with magnetometer.**



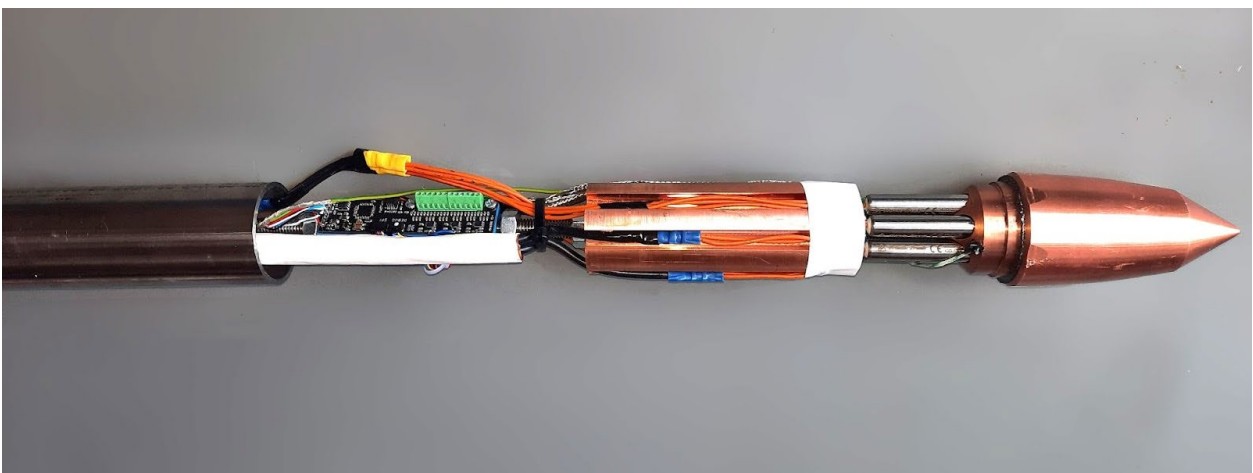


**Figure 4 – Overview of the internal construction of the melt tip before resin casting.**


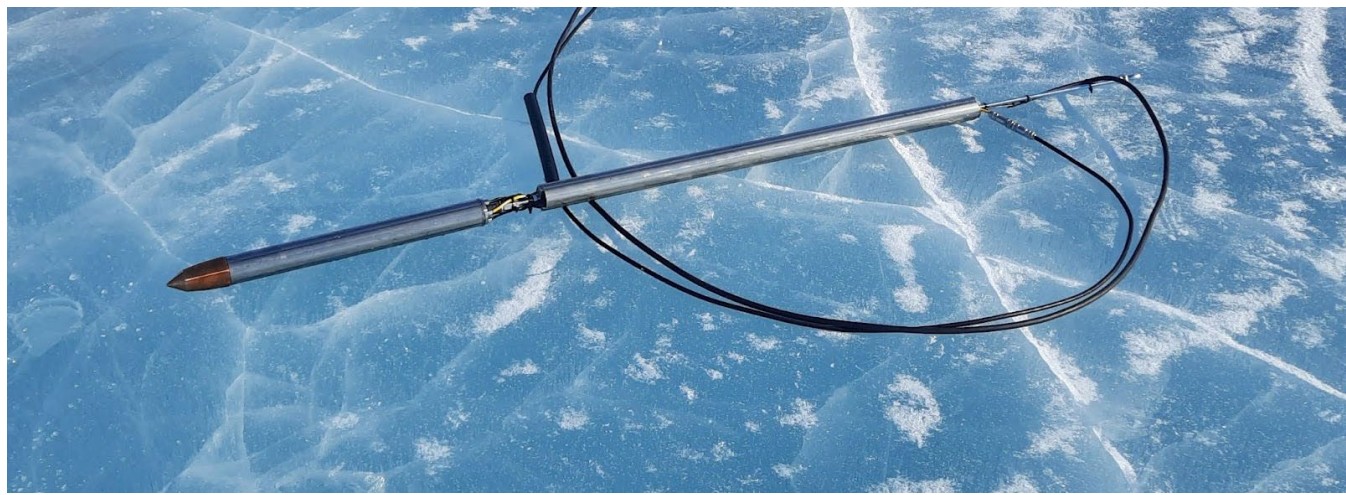

**Figure 5 – Photo of a fully assembled v3 melt-tip on lake ice. The lower member (image left) is sealed and resin-filled while the upper member (image right) is unsealed and unfilled.**





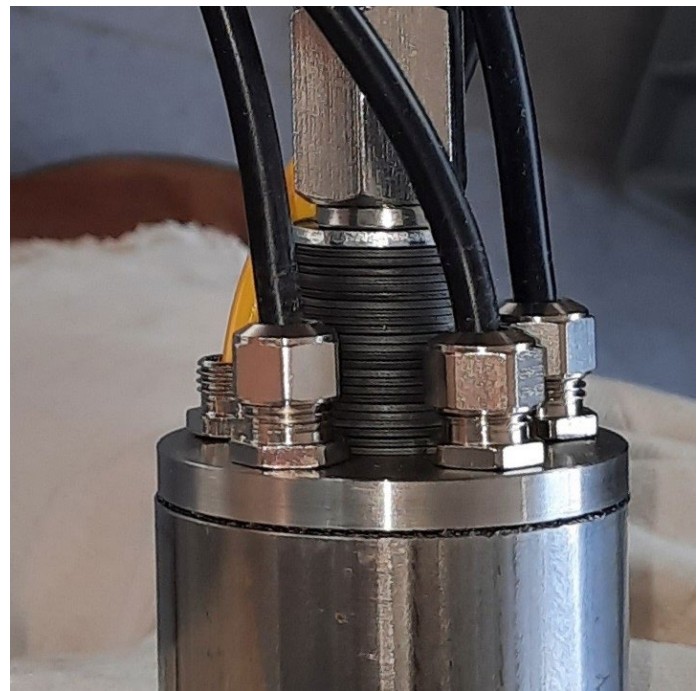

**Figure 6 – With four power cables, one data cable and an M10 threaded bolt covered in Belleville washers, there is**
**very little free space on the metal top cap of the lower member of the melt tip.**


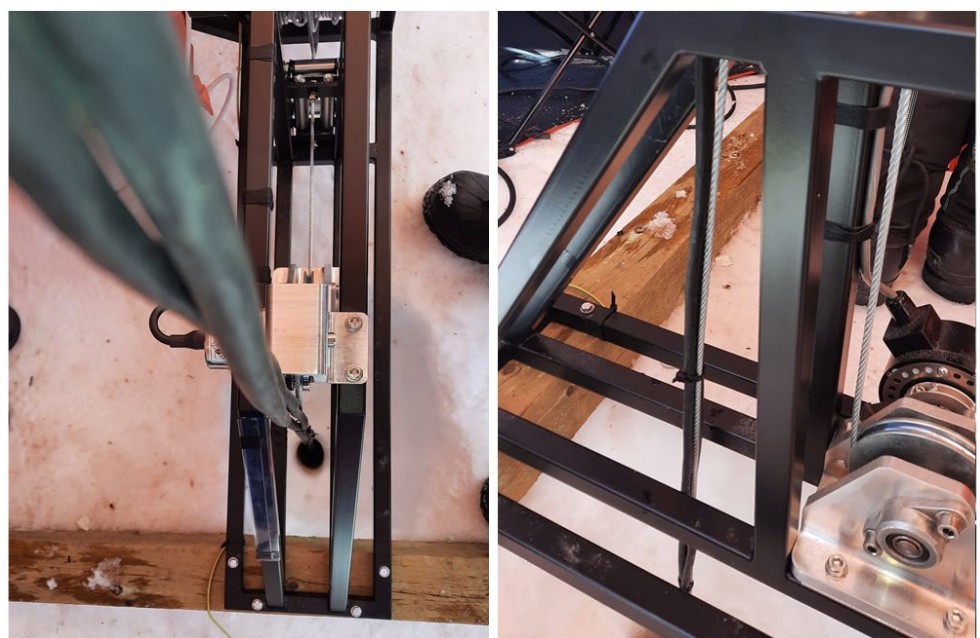

**Figure 7 – Overhead (Left) and side (Right) views of the umbilical cable bundled with zip ties prior to entering the borehole.**






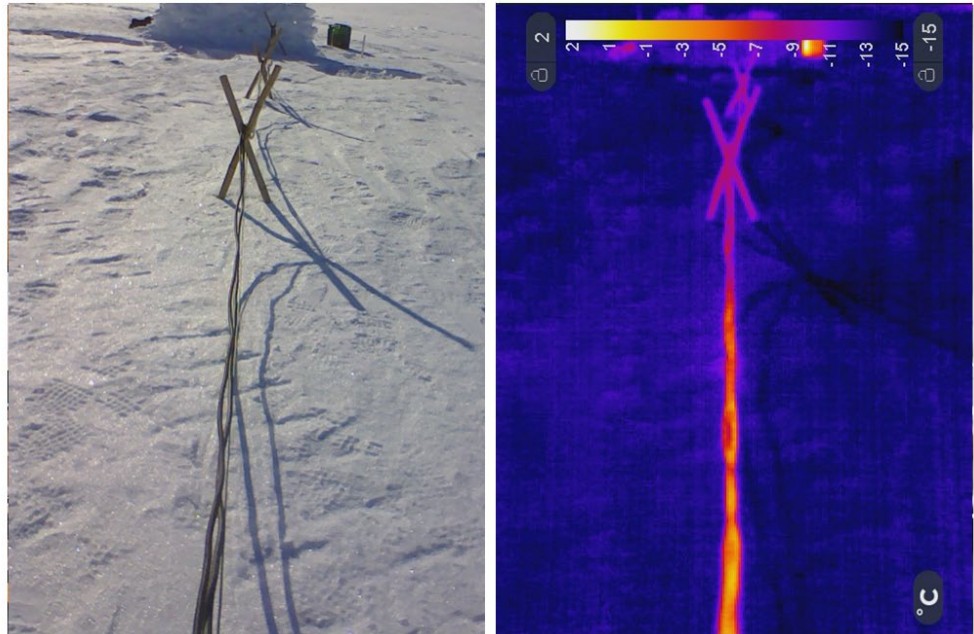

**Figure 8 – The effect of resistive heating of power cables during field testing. Left: Visible image of the 230 and 400 V power cables from the generator to the drilling tent. Right: Thermal image of the same cables highlights that current flow warms the cable to 0 °C, despite an ambient air temperature of -10 °C with light winds.**






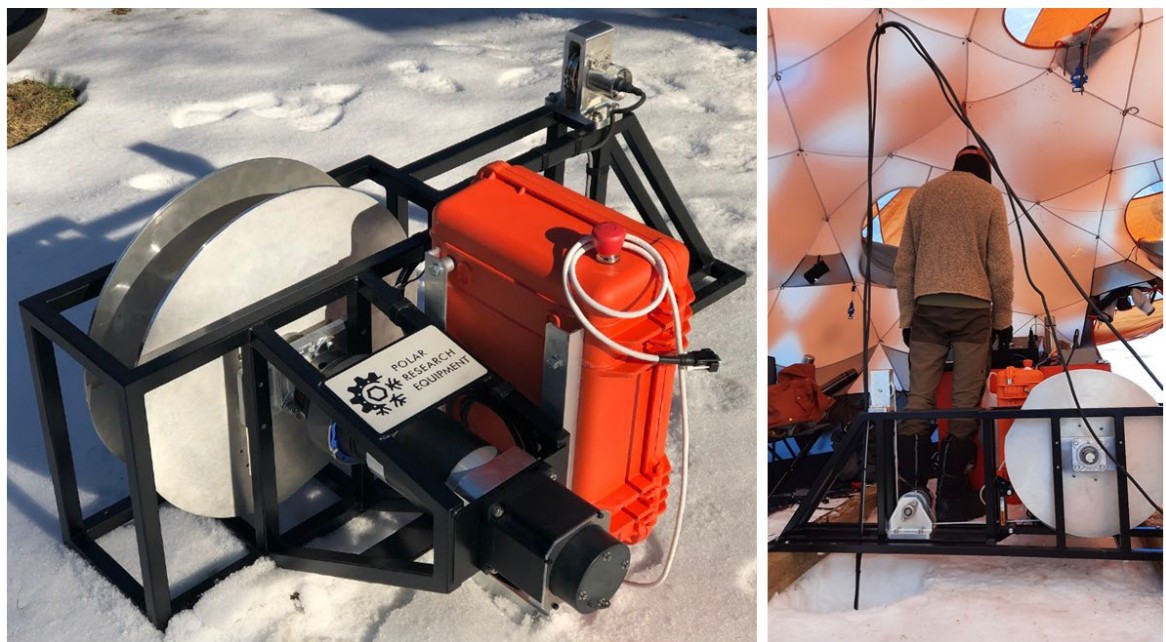

**Figure 9 – Left: Winch motor-side view prior to field testing. Right: Winch spool-side view during field testing. Umbilical components pass through a carabiner on the tent ceiling before dropping into the winch frame and being zip-tied to the structural wire.**






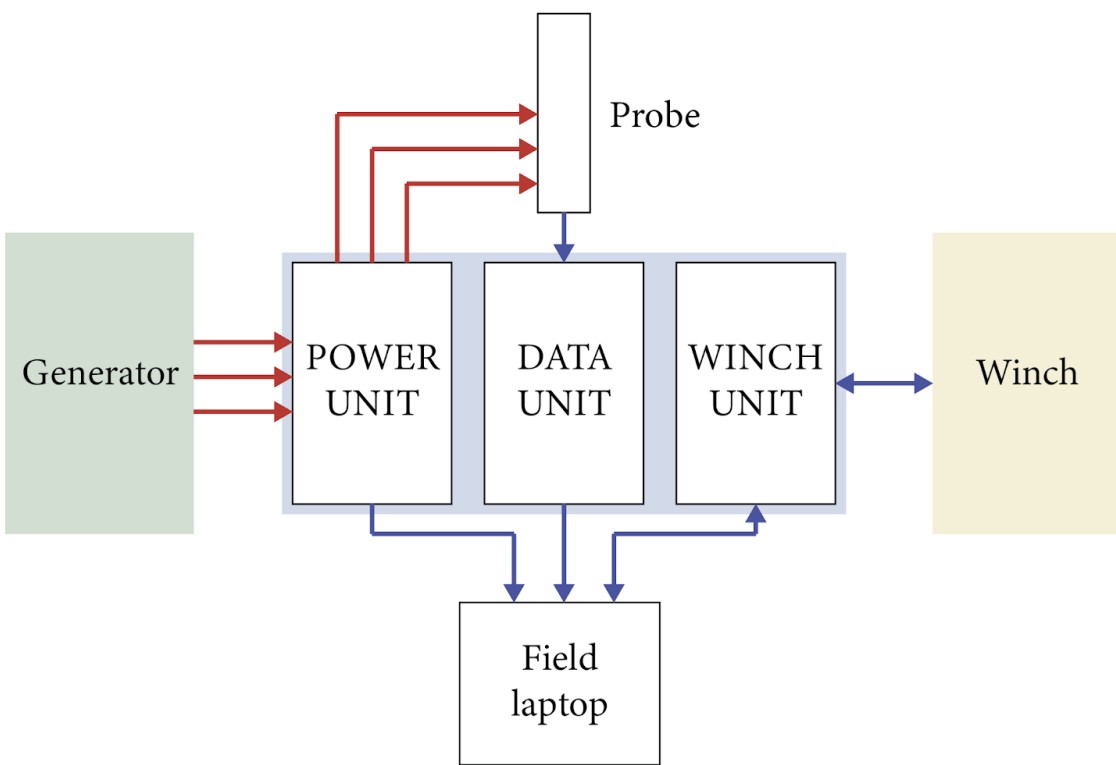

**Figure 10 – Diagram of the interface connections to other drilling system components. Red arrows indicate power.**
**Blue arrows indicate data.**




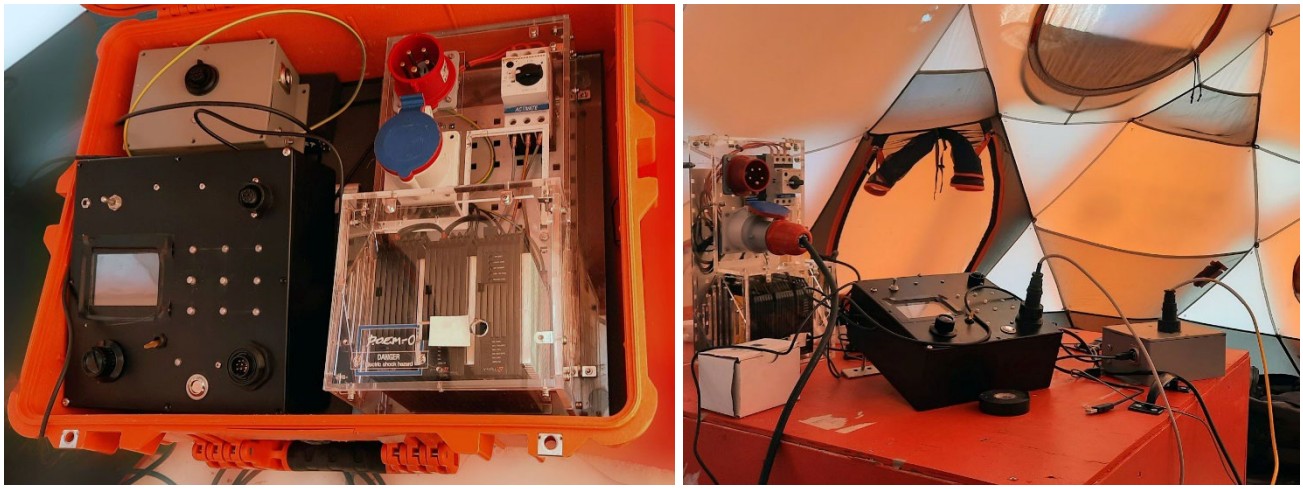

**Figure 11 – Interface connecting to the melt-tip, winch, and power supply at the ice-sheet testing site. Left: Interface elements transported inside a Pelican case. Right: Interface elements connected in the field tent.**








**Figure 12 – Sample of melt-tip data feedback during ice-sheet drilling on 13 May 2022. Top to Bottom: temperatures in four cartridge heaters; temperatures in two spots in the copper heating block; temperature at three places on the electronics board; 3-D acceleration; roll and pitch.**




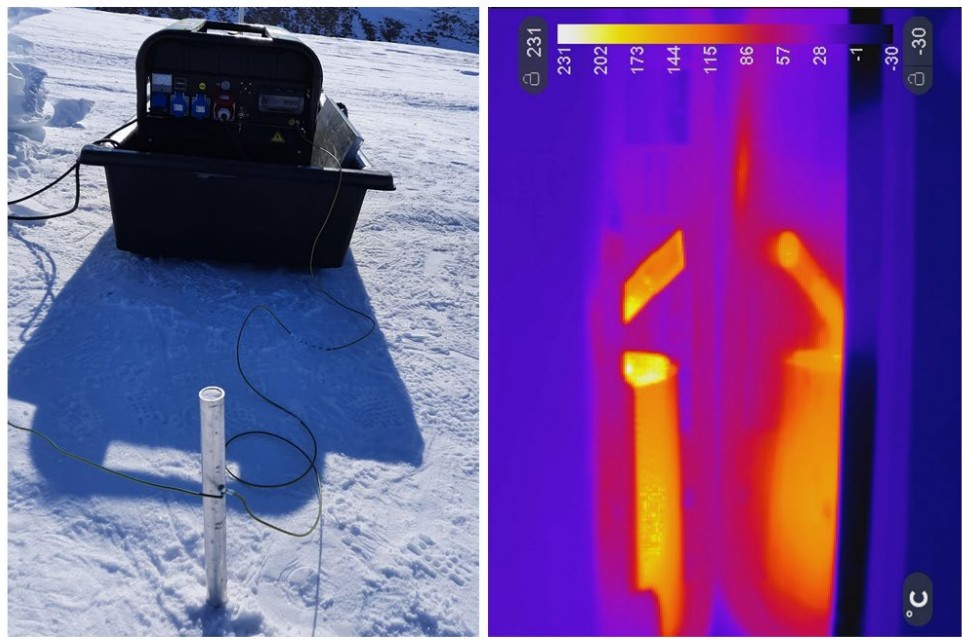

**Figure 13 – Left: Grounding stake drilled into the ice sheet near the generator to ground all electrical elements of the drilling system. Note the metal heat deflector beside the generator, protecting the high-density plastic snowmobile sled. Right: Thermal image of the generator exhaust pipe and its reflection in the metal heat deflector.**






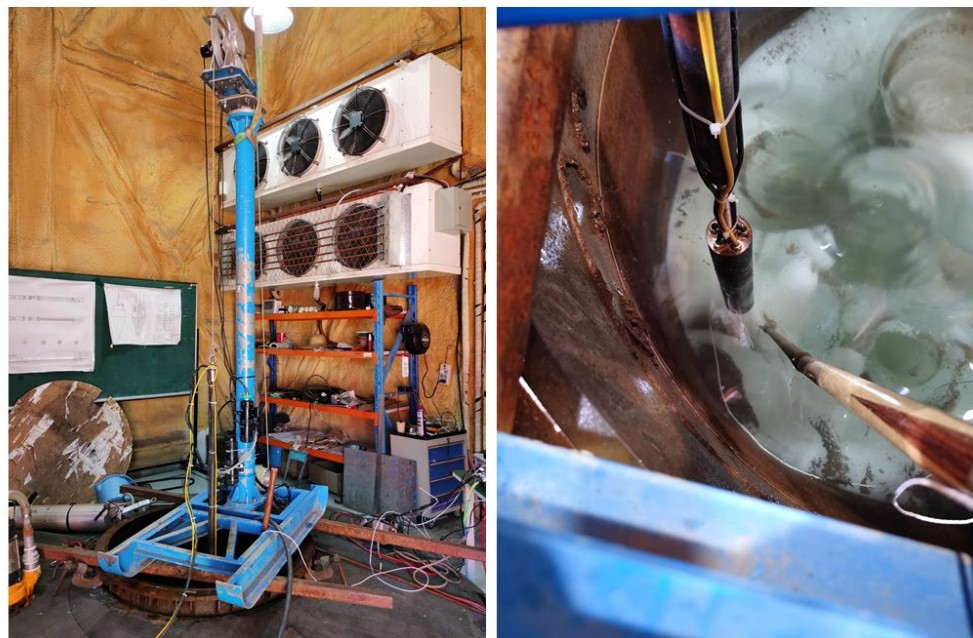

**Figure 14 – Left: The melt tip is attached to a 2 m drill flight and suspended from an oversized winch above the ice well during laboratory testing. Right: Looking down into the ice well as the melt tip starts a new borehole. A matrix of previous boreholes is seen in the ice well.**




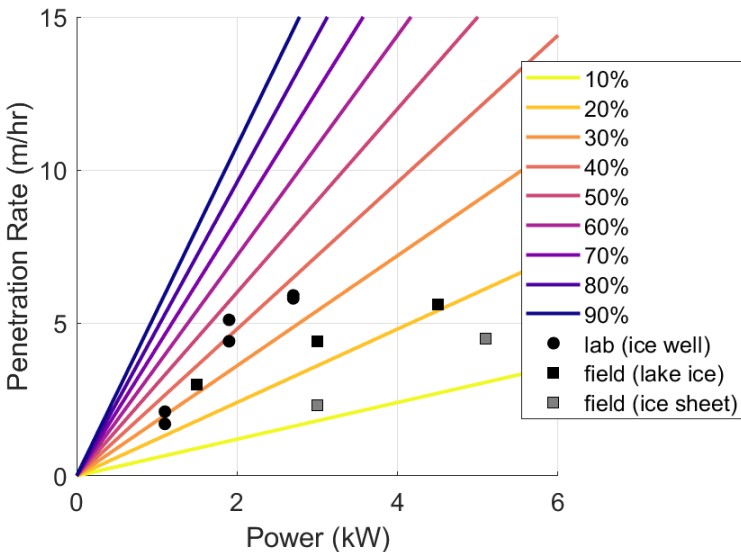

**Figure 15 – Penetration rate as a function of power for the v2 melt tip in the ice well and the v3 melt tip in the lake ice and ice sheet. For the ice sheet 21 m test, the 1-hour maximum is shown in addition to the test mean. Colored lines indicate heat transfer efficiency in comparison to theoretical perfect heat transfer from the cartridge heaters to melt**
**an ideal borehole.**



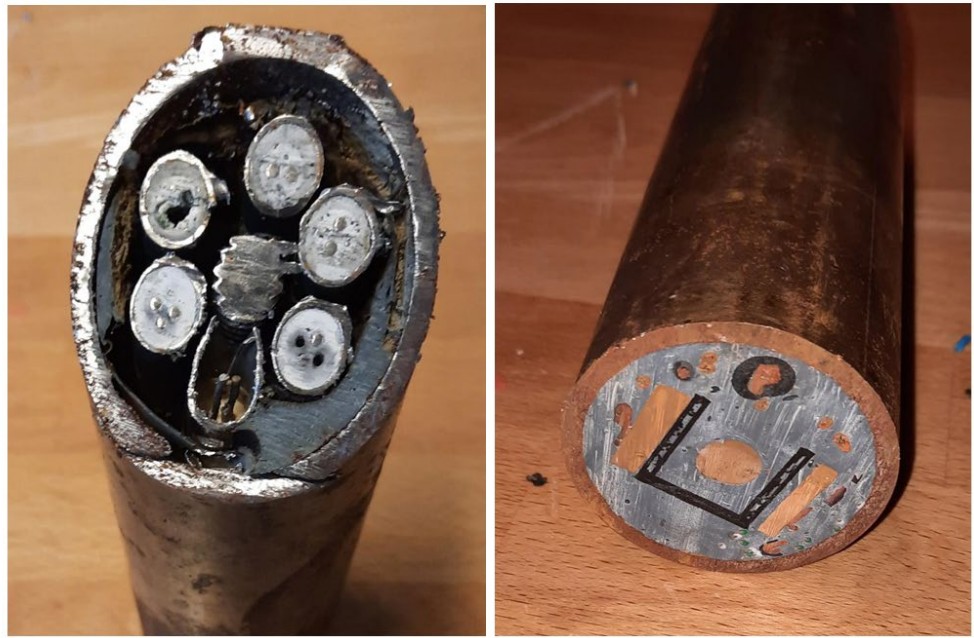

**Figure 16 – Melt-tip cross sections from destructive testing of the v2 melt tip. Left: The resin filling encasing the unheated portions of the cartridge heaters resin exposed from the copper heating block has melted. Right:**
**Uncompromised resin filling higher in the melt tip, encasing the electronics package**.



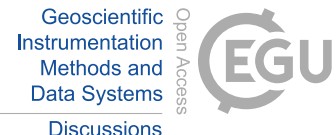

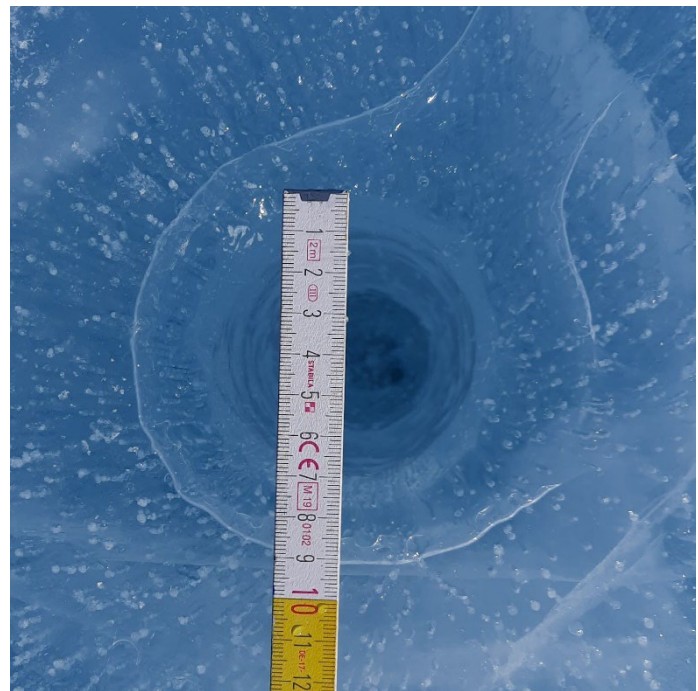

**Figure 17 – Testing on the frozen Lake Tuto suggests that the effective borehole diameter of the melt tip is ~70 mm over a wide range of power levels and penetration rates.**






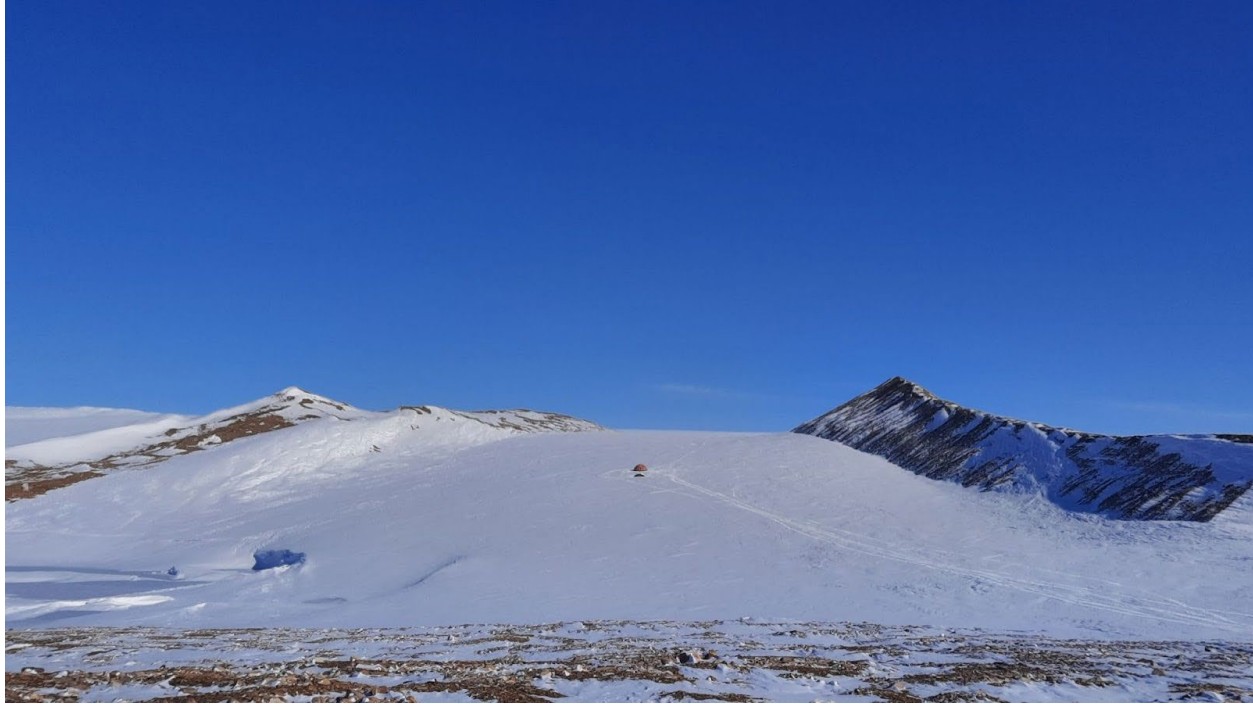

**Figure 18 – The D-11 drill site located on Tuto Ramp, near Thule Air Base, in May 2022. In 1961, the ice surface elevation at D-11 was approximately equal to the current elevations of the ramp roads located both north (image left) and south (image right) of the drill tent.**






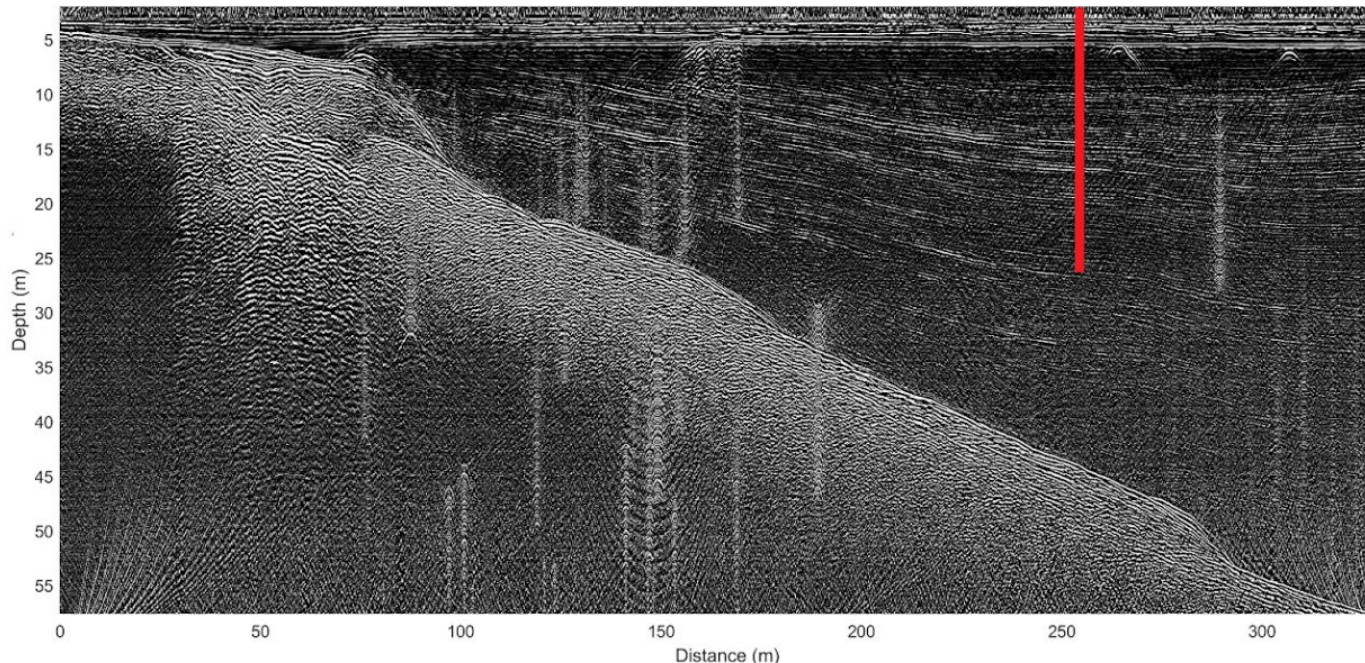

**Figure 19 – 100MHz ice-penetrating radar of ice thickness from the ice-sheet margin inland past the D-11 borehole (denoted in red) suggests an ice thickness of ~44 m at the D-11 drill site.**






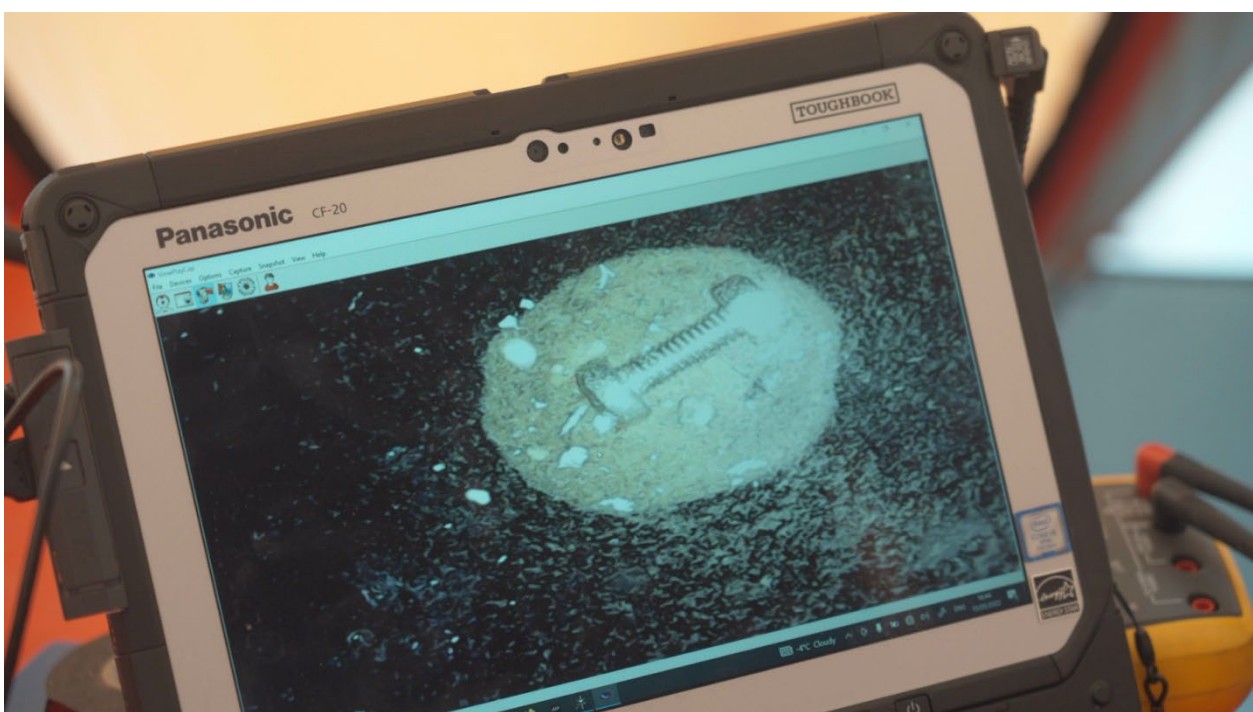

**Figure 20 – Down-borehole video camera shows substantial sediment accumulation on the borehole bottom within 5 m of the ice surface. We estimate that a water-filled cavern of diameter ~50 cm formed around the melt tip. A bolt of length ~1.5 cm was dropped into the frame for scale.**



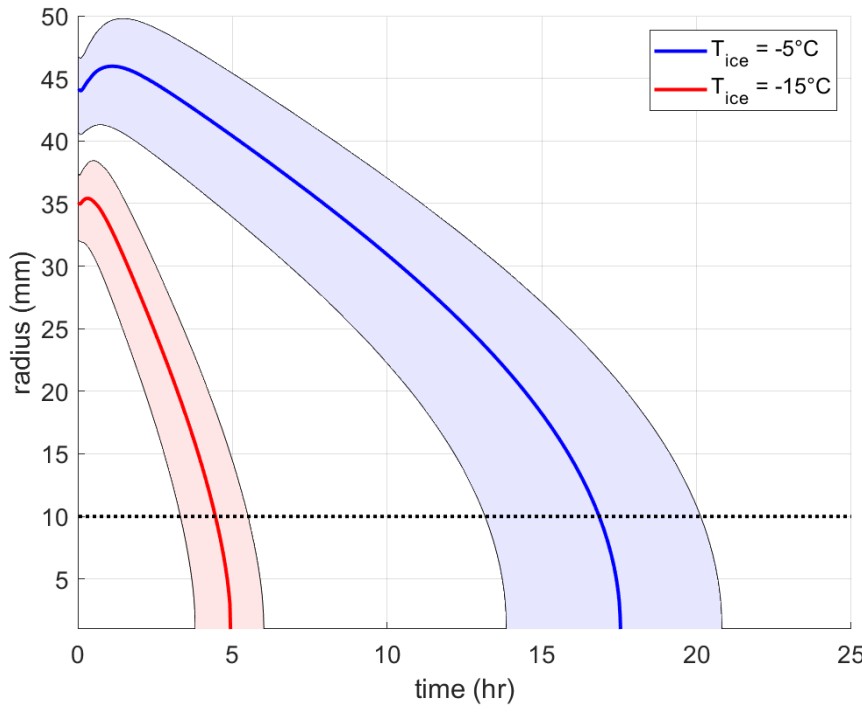


**Figure 21 – Borehole radius over time solved with a 1-D ice-water moving boundary heat diffusion model. These contrasting simulations depict the differing maximum borehole radii and decay timescales with ice temperatures of -5 and -15 °C. Shaded areas denote 75±25 °C uncertainty in initial borehole water temperature. Dotted line denotes a 10 mm radius equivalent to the effective cross-sectional area of the umbilical**.






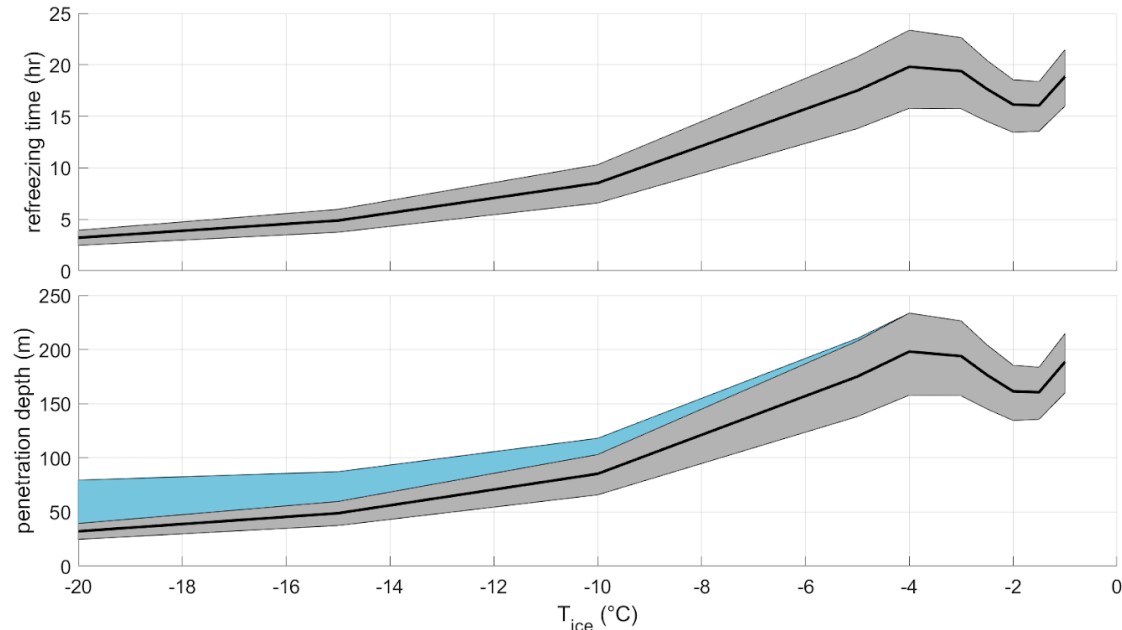

**Figure 22 – Top: Borehole refreezing time to a characteristic water radius of 1 mm over a variety of ice temperatures. Shaded area denotes 75±25 °C uncertainty in initial borehole water temperature. Bottom: Maximum theoretical penetration depth of the ice drilling system over a variety of ice temperatures. Grey shaded area corresponds to the shaded area in the top subplot. Blue shaded area reflects the characteristic porous firn depth.**


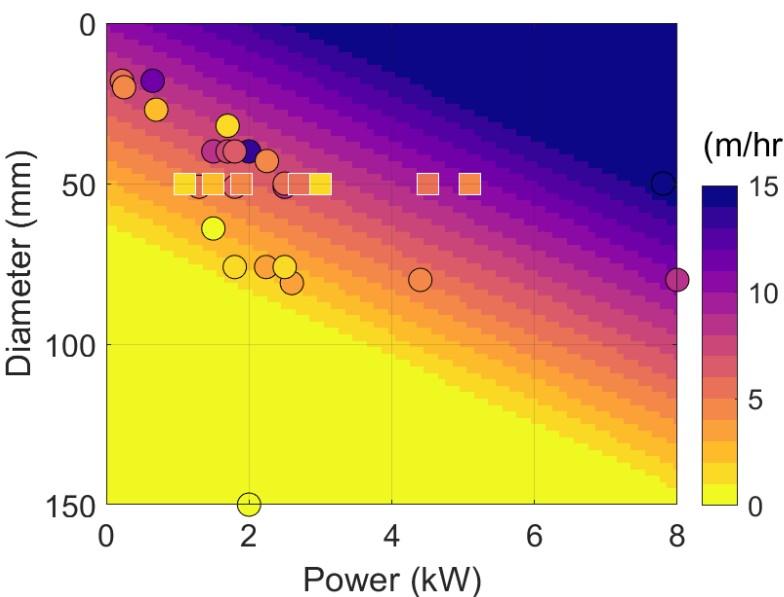

**Figure 23 – Rate of penetration as dependent on melt-tip diameter and power. Circles denote previously described melt tip statistics compiled by Talalay [2019]. Background color denotes the expected rate of penetration based on bivariate regression of these previously described melt tip statistics. Squares denote our melt tip performance under both laboratory and field conditions described here.**