# Peer review of "Design and Performance of the Hotrod Melt-Tip Ice-Drilling System"

_Geoscientific Instrumentation, Methods and Data Systems, 2022_

## Author Response (AR1)

**Reviewer 1:**

This paper presents a very detailed account of design, together with very limited engineering data from laboratory and field tests, for a 2 m long, 5 cm diameter thermal ice drill intended to emplace thermistor strings vertically in ice sheets and glaciers. Electrical power (typically 1-6 kW) is supplied by a gasoline generator on the ice surface and conveyed to the drill via a tether paid out from the surface. The drill differs in various particulars (e.g., in using custom-made electrical cartridge heaters to try to direct heat preferentially along one axis), but is not fundamentally different in concept from many predecessors of similar size described in the literature.

The laboratory test data comprise 5-6 runs in pure ice at constant powers of 1.1-2.7 kW to depths of ~2 m in a 1 m diameter ice column at approximately -10 C (cf. pg. 12 and Fig. 2). Initial field tests are reported to "< 1 m" depth in lake ice near Thule Air Base in Greenland, ice temperature unspecified, at power levels 1.5-4.5 kW, with corresponding descent rates of 3-5.6 m/hr (Figure 2). A melt-hole diameter of 7 cm in the lake ice is reported (in apparent contrast to the laboratory cases). Two probe runs are also reported in an ice sheet ablation area near Thule at low but unspecified elevation, where ice was perhaps as thick as 44 m. No information on ice temperature, neither near-surface nor versus depth, is given. The first run reached 5 m depth over 3 hours using 3 kW of power, but was arrested by an accumulation of sediment at the bottom of the melt hole. The second run, initiated 2 m laterally distant from the first, reached 21 m depth over ~9 hours using 4.2 kW of power, before encountering a sediment layer which may have been detected independently by radar at that depth. There is no mention of melt-hole diameter in these latter two runs.

In the course of reviewing this paper, I re-read a number of literature accounts of lightweight thermal drilling efforts in the past, including Nizery (1951), LaChapelle (1963), Classen (1967), Gillet (1975), Taylor (1976), Rado et al. (1987), Kelley et al. (1994), and German et al. (2021) (none of which are referenced by the authors), as well as Dachwald et al. (2014), Zagorodnov et al. (2014) and Heinen et al. (2021) (which the authors do reference, albeit incorrectly in the case of Heinen et al.). These accounts all provide more detailed test results for ice penetration, to greater depths or (in the case of German et al.) with greater scientific return, than is the case in this paper.

I am therefore presently without a clear, compelling answer to the question of what scientific or engineering contribution this paper adds to the existing literature. (A detailed design alone for a probe not shown to offer any new capability does not qualify, in my view.) Neither is this question addressed in the paper so far as I can see. I would be open to an argument for what such a contribution could be, but absent such an argument at present, I am unable to recommend this paper for publication.

References

Classen, D.F. (1967), "Thermal Drilling and Deep Ice-Temperature Measurements on the Fox Glacier, Yukon", M.S. Thesis, University of British Columbia.

*German, L., J.A. Mikucki, S.A. Welch, K.A. Welch, A. Lutton and B. Dachwald (2021), "Validation of sampling antarctic subglacial hypersaline waters with an electrothermal ice melting probe (IceMole) for environmental analytical geochemistry", International Journal of Environmental Analytical Chemistry 101(15), 2654-2667.*

*Gillet, F. (1975), "Steam, Hot-Water and Electrical Thermal Drills for Temperate Glaciers", Journal of Glaciology 14(70), 171-179.*

*Kelley, J.J., K. Stanford, B. Koci, M. Wumkes,  and V. Zagorodnov (1994), "Ice Coring and Drilling Technologies Developed by the Polar Ice Coring Office", Memoirs of National Institute of Polar Research Special Issue 29, 24-40.*

*LaChapelle, E. (1963), "A Simple Thermal Ice Drill", Journal of Glaciology 4(35), 637-642.*

*Nizery, A. (1951), "Electrothermic Rig for the Boring of Glaciers", Eos Transactions of the American Geophysical Union 32(1), 66-72.*

*Rado, C., C. Girard, and J. Perrin (1987), "Electrochaude: A Self-Flushing Hot-Water Drilling Apparatus for Glaciers with Debris", Journal of Glaciology 33(114), 236-238.*

*Taylor, P.L. (1976), "Solid-Nose and Coring Thermal Drills for Temperate Ice", pgs. 166-176 in Ice Core Drilling, J.F. Slpettstoesser, ed., University of Nebraska Press.*

We thank Reviewer 1 for their summary our melt-tip ice-drilling system and its testing. We do not state that our drilling system is fundamentally different from predecessor systems. However, the specific power density of our system that we now present (305 W/cm$^2$) is approximately twice the maximum specific power density of any existing melt-tip drill deployed in the past fifty years [Talalay, 2019]. We now more explicitly highlight this very high specific power density from existing melt-tip systems. We also more thoroughly cite the relevant historical literature that has been suggested by Reviewer 1.

In the revised manuscript, we also provide a better description of field test sites. This includes the coordinates and elevation of the D-11 ice-sheet borehole (76.4106°N, 68.2876°W, 528 m), as well as the coordinates and elevation of the lake boreholes (76.4124°N, 68.2949°W, 496 m). We also specify the approximate borehole ice temperatures as being -10°C, based on ice temperatures observed during the drilling period at 8 m depth at the THU_L PROMICE automatic weather station located <1 km away [Fausto et al., 2021].

We acknowledge that borehole diameter is an important parameter. Unfortunately, our melt-tip cannot measure borehole diameter, so we cannot provide further information on borehole diameter during testing beyond photography at the lake ice testing site. We now state this explicitly. The boreholes in the artificial ice well were too recessed within the ice well to allow similar photography. The ice-sheet boreholes were obscured by ~1.5 m of snow cover, which similarly prevented measuring diameter from overhead photographs.

While our manuscript does not present novel scientific findings, we feel that publishing an open-access design for our melt tip and ancillary elements falls within the scope of

*Geoscientific Instrumentation, Methods and Data Systems*, as it allows other research groups to broadly benefit from our design and testing. For example, the Colgan et al. [2022] data repository associated with this manuscript contains, what we believe is the first open-access numerical code for borehole refreezing via a radial enthalpy solution. Novel scientific findings resulting from our melt-tip ice-drilling system will be published, in time.

Colgan, W., Shields, C., Lines, A., Elliot, J., and Rajaram, H. Hotrod melt-tip ice-drilling system. https://doi.org/10.22008/FK2/DXXR06. GEUS Dataverse, V1. 2022.

Fausto, R. S., van As, D., Mankoff, K. D., Vandecrux, B., Citterio, M., Ahlstrøm, A. P., Andersen, S. B., Colgan, W., Karlsson, N. B., Kjeldsen, K. K., Korsgaard, N. J., Larsen, S. H., Nielsen, S., Pedersen, A. Ø., Shields, C. L., Solgaard, A. M., and Box, J. E.: Programme for Monitoring of the Greenland Ice Sheet (PROMICE) automatic weather station data, Earth Syst. Sci. Data, 13, 3819–3845, https://doi.org/10.5194/essd-13-3819-2021, 2021.

Talalay, P. Hot-Point Drills. In: Thermal Ice Drilling Technology. 1-80. Springer Geophysics. Springer. https://doi.org/10.1007/978-981-13-8848-4_1, 2019.

**Reviewer 2 (Kris Zacny):**

*Very informative and detailed paper. It's always great to see the new advancements!*

We thank Kris Zacny for summarizing our manuscript as informative and detailed.

**Reviewer 3:**

*This manuscript is a detailed technical description of a light weighted drill. I am very interested in the development of this type of electro thermal ice-drilling systems and new ideas within this area. I like that the authors are willing to share their development in an open-access repository.*

*My main critic is that I'm missing new concepts, ideas within the scope of this development and manuscript. Several electro thermal ice-drilling systems where developed and used over the last decades. The coauthor Pavel Talalay summarized several developments in his book "Thermal Ice Drilling Technology" (Springer, 2019). Please point out, what are the substantial new concepts, ideas, methods, or data within this manuscript.*

*In addition, I am still missing some smaller information on the design decisions and subsystem information. Please discuss how you calculated the values for your expected speed/penetration rate. Also, the specific power density of our melting tip is not included and discussed. In the Jilin laboratory test you recorded very detailed data and you present*

*only mean values, without a discussion. For the description of the field tests I'm missing environmental parameters for the test in field, e.g. temperatures of the ice and the ice density.*

We thank Reviewer 3 for the value that they place on both detail and open-access design plans.

In the revised manuscript, we more thoroughly compare our drilling system to similar existing systems. Briefly, aside from providing an open-access design, the main difference of our system is its relatively high power. With an idealized cross-sectional area of 19.6 $cm^2$ (equivalent to a circle of radius 2.5 cm) our drill provides a specific power density of 306 $W/cm^2$ at 6 kW power. We now explicitly present this specific power density.

Most melt-tip drills of similar cross-sectional area, especially those under development for extraterrestrial applications, are powered with a small fraction of this specific power density. Partly as a consequence of accommodating larger than normal power cables, our winch is substantially more robust, and well-documented, than in most previous systems.

While we defended our derivation of the expected rates of penetration shown in discussion version Figure 23 in the GI Discussion forum (https://doi.org/10.5194/gi-2022-18-AC4), on further reflection, we have removed explicit visualization of this bivariate regression (revised Figure 24). In now give better coverage of previous melt-tip performances, and we merely say that the data compiled by Talalay [2019] "…suggests that penetration rate increases ~1.5 m/hr for every 1 kW increase in system power, and that penetration rate conversely decreases ~1.5 m/hr for every 1 cm increase in melt tip diameter."

While the artificial ice well tests at Jilin University did record penetration rate each second, there was very little temporal variation around the mean penetration rate. For example, during the 45% power test shown below, the penetration rate was 5.9 ± 0.3 m/hr (between 25 and 150 cm depth; revised Figure 15). This was similar across all ice-well tests. We therefore only discuss the mean rate of penetration for each test. We now state this explicitly in the revised manuscript.

Finally, with regards to the field site, we do not have measurements of ice density with depth at the drill site, but we can assume that the ablation zone ice has a bulk density, similar to that of pure ice. We can, however, add that ice temperature at the drill site was approximately -10°C, based on ice temperatures observed during the drilling period at 8 m depth at the THU_L PROMICE automatic weather station located <1 km away [Fausto et al., 2021]. We now state this explicitly.

Fausto, R. S., van As, D., Mankoff, K. D., Vandecrux, B., Citterio, M., Ahlstrøm, A. P., Andersen, S. B., Colgan, W., Karlsson, N. B., Kjeldsen, K. K., Korsgaard, N. J., Larsen, S. H., Nielsen, S., Pedersen, A. Ø., Shields, C. L., Solgaard, A. M., and Box, J. E.: Programme for Monitoring of the Greenland Ice Sheet (PROMICE) automatic weather station data, Earth Syst. Sci. Data, 13, 3819–3845, https://doi.org/10.5194/essd-13-3819-2021, 2021.

Talalay, P. Hot-Point Drills. In: Thermal Ice Drilling Technology. 1-80. Springer Geophysics. Springer. https://doi.org/10.1007/978-981-13-8848-4_1, 2019.

---

## Author Response (AR2)

Dear Andy, thank you for securing further review of our manuscript and inviting minor revisions.

Re #1 – placing our work in a quantitative context – we have now included introduction Figure 1, which highlights the specific power density of all hot point drill developed to date. We have color coded this figure to highlight the increasing prevalence of low power density hot points intended for extraterrestrial settings. This figure clearly shows that no other groups have been experimenting with >200 W/cm2 specific power since c. 1980. Our Hotrod drilling system is quite peerless in terms of the specific power systems of all other hot points in operation today.

Re: #2 – relation of manuscript to digital assets – we have now included text stating that in addition to providing technical specifications, the article supplements digital assets by providing the rationale behind design choices, outlining abandoned variants and failures, describing digital data and software solutions, and highlighting outstanding challenges. We specifically state that while hot points of similar power density were deployed in the 1950s, there are no detailed designs of these drills in the public sphere today.

Re: #3 – Nizery [1951] specific power– We have now calculated the specific power of all n = 46 hot points surveyed by Talalay [2019]. We assess a specific power of 397 W/cm2 to the Nizery [1951] design. We also clarify our statement that the specific power of our hot point is twice the specific power of any other hot point in operation since c. 1980. This clarifies that there are no drills currently operating at this specific power density, not that our specific power is the highest ever designed (which is clearly Nizery [1951]).

Thank you for your editorial service in support of open science.